# Novel Set of Diarylmethanes to Target Colorectal Cancer: Synthesis, *In Vitro* and *In Silico* Studies

**DOI:** 10.3390/biom13010054

**Published:** 2022-12-27

**Authors:** Ameni Hadj Mohamed, Aline Pinon, Nathalie Lagarde, Elizabeth Goya Jorge, Hadley Mouhsine, Moncef Msaddek, Bertrand Liagre, Maité Sylla-Iyarreta Veitía

**Affiliations:** 1Laboratoire Génomique, Bioinformatique et Chimie Moléculaire (GBCM, EA 7528) Conservatoire National des Arts et Métiers, HESAM Université, 2 rue Conté, 75003 Paris, France; 2Laboratoire de Chimie Hétérocyclique, Produits Naturels et Réactivité (LR11ES39) Université de Monastir Avenue de l’Environnement, Monastir 5019, Tunisia; 3Univ. Limoges, LABCiS, UR 22722, Faculté de Pharmacie, F-87000 Limoges, France; 4Department of Food Sciences, Faculty of Veterinary Medicine, University of Liège, Av. de Cureghem 10 (B43b), 4000 Liège, Belgium; 5Peptinov, Pépinière Paris Santé Cochin, Hôpital Cochin, 29 rue du Faubourg Saint Jacques, 75014 Paris, France

**Keywords:** diarylmethane, colorectal cancer, antiproliferative activity, apoptosis, molecular docking, ADME profile

## Abstract

Distinctive structural, chemical, and physical properties make the diarylmethane scaffold an essential constituent of many active biomolecules nowadays used in pharmaceutical, agrochemical, and material sciences. In this work, 33 novel diarylmethane molecules aiming to target colorectal cancer were designed. Two series of functionalized olefinic and aryloxy diarylmethanes were synthesized and chemically characterized. The synthetic strategy of olefinic diarylmethanes involved a McMurry cross-coupling reaction as key step and the synthesis of aryloxy diarylmethanes included an O-arylation step. A preliminarily screening in human colorectal cancer cells (HT-29 and HCT116) and murine primary fibroblasts (L929) allowed the selection, for more detailed analyses, of the three best candidates (**10a**, **10b** and **12a**) based on their high inhibition of cancer cell proliferation and non-toxic effects on murine fibroblasts (<100 µM). The anticancer potential of these diarylmethane compounds was then assessed using apoptotic (phospho-p38) and anti-apoptotic (phospho-ERK, phospho-Akt) cell survival signaling pathways, by analyzing the DNA fragmentation capacity, and through the caspase-3 and PARP cleavage pro-apoptotic markers. Compound **12a** (2-(1-(4-methoxyphenyl)-2-(4-(trifluoromethyl)phenyl) vinyl) pyridine, Z isomer) was found to be the most active molecule. The binding mode to five biological targets (i.e., AKT, ERK-1 and ERK-2, PARP, and caspase-3) was explored using molecular modeling, and AKT was identified as the most interesting target. Finally, compounds **10a, 10b** and **12a** were predicted to have appropriate drug-likeness and good Absorption, Distribution, Metabolism and Excretion (ADME) profiles.

## 1. Introduction

Diarylmethanes (DAM) are a class of organic compounds whose chemical scaffold is based on a central methylene group and a 1,1-diaryl unit. These molecules from either natural or synthetic source constitute attractive scaffold present in various pharmaceutical agents. DAM are privileged structures in medicinal chemistry. They endowed several biological activities, including anti-infectious [1,2], anti-inflammatory [3,4,5] and anti-cancer activity [6,7,8,9,10,11,12,13,14,15,16,17]. Moreover, some DAM are well known as therapeutical agents [18,19,20,21,22,23,24,25]. The most relevant example is tamoxifen which has been successfully used for decades in the treatment of hormone-dependent breast cancer [17,18,19] (Figure 1).

Colorectal cancer (CRC) is the third most common cancer-related death after prostate and lung cancer in western societies, with more than half a million annual deaths worldwide. The incidence is high and is steadily increasing. In France, CRC is the second leading cause of death (17,117 deaths/year) [26,27]. The current therapies used in the clinic of CRC still have several side effects and resistance issues. Therefore, there is an urgent need to find novel therapeutical alternatives to reduce these drawbacks.

We described previously the synthesis of ferrocenyl DAM using McMurry cross coupling and the antibacterial activity evaluation [2,28]. Inspired by this study, and by the evidence that endows that DAM are well described for their anti-cancer activities, we were interested in the design and synthesis of new pyridyl diarylmethanes-type to develop novel therapeutic agents to target CRC.

Hence, the pyridyl DAM olefinic and their oxygenated analogues were synthesized, and their antiproliferative activity was assayed in vitro on human CRC cell lines. The potential cytotoxicity of the synthesized compounds was evaluated on normal murine fibroblast. Several target receptors involved in the inhibition of cancer cell proliferation were analyzed in vitro to suggest the likely mechanism of action of the DAM drug candidates. Moreover, we complemented our endeavors with a docking study performed on some biological targets that may potentially be involved in the biological activity. Finally, an *in silico* prediction of biopharmaceutical properties allowed us to investigate the Absorption, Distribution, Metabolism, and Excretion (ADME) profile and druglikeness of the most promising anticancer DAM.

## 2. Materials and Methods

### 2.1. Chemistry

All reagents were obtained from commercial sources unless otherwise noted and used as received. Heated experiments were conducted using thermostatically controlled heating mantles and were performed under an atmosphere oxygen-free in oven-dried glassware when necessary. The reactions were monitored by analytical Thin Layer Chromatography (TLC). TLC was performed on aluminum sheets precoated silica gel plates (60 F254, Merck). TLC plates were visualized using irradiation with light at 254 nm. Flash column chromatography (FCC) was carried out when necessary, using silica gel 60 (particle size 0.040–0.063 mm, Merck). A mixture of cyclohexane (CyHex) and ethyl acetate (EtOAc) was used as mobile phase.

### 2.2. Physical Measurements

Melting points were recorded on a Kofler hot block Heizbank type 7841 and were uncorrected. The structures of the products were checked by comparison of Nuclear Magnetic Resonance (NMR), Infrared (IR) and Mass spectrometry (MS) data and by the TLC behavior. ^1^H- and ^13^C-NMR spectra were recorded on a Bruker BioSpin GmbH spectrometer 400 MHz, at room temperature. Chemical shifts are reported in *δ* units, parts per million (ppm). Coupling constants (*J*) are measured in hertz (Hz). Splitting patterns are designed as follows: s, singlet; d, doublet; dd, doublet of doublets; m, multiplet; t, triplet; td, triplet of doublet; ddd: doublet of doublet of doublet. Distortionless enhancement with polarization transfer (DEPT) experiments and various 2D techniques such as COrrelation SpectroscopY (COSY), Heteronuclear Single Quantum Coherence (HSQC) and Heteronuclear Multiple Bond Correlation (HMBC) were used to establish the structures and to assign the signals. Conventional adopted to assign signal of ^1^H- and ^13^C-NMR spectra are described in Figure 2. Gas chromatography-mass spectrometry (GC-MS) analysis was performed with an Agilent 689 0N instrument equipped with a dimethyl polysiloxane capillary column (12 m × 0.20 mm) and an Agilent 5973N MS detector-column temperature gradient 80–300 °C (method 80): 80 °C (1 min); 80 °C to 300 °C (12.05 °C/min); 300 °C (2 min). Electrospray ionization (ESI)-Low resolution mass spectra (LRMS) were performed from ionization by electrospray on a Waters Micromass ZQ2000. Infrared spectra were recorded over the 400–4000 cm^−1^ range with an Agilent Technologies Cary 630 Fourier-transform infrared spectroscopy (FTIR)/Attenuated Total Reflectance (ATR)/ZnSe spectrometer. High-resolution mass spectra (HRMS) analyses were acquired on a Thermo Scientific LTQ Orbitrap mass spectrometer.

### 2.3. General Procedure for the Synthesis of Olefinic Diarylmethanes

To a suspension of zinc (2.86 mmol, 182 mg, 6 eq) in anhydrous tetrahydrofuran (THF) (2.6 mL) under argon was added dropwise TiCl_4_ (1.872 mmol, 0.2 mL, 4 eq). The reaction mixture was stirred for 2 h at 85 °C. A solution of and (4-methoxyphenyl) (pyridin-2-yl)methanone **5** (0.468 mmol, 100 mg, 1 eq) and the corresponding aromatic aldehyde **2** (0.486 mmol, 1.04 eq) in THF (1 mL) was then added dropwise via a syringe. After reaction completion, the mixture was cooled at room temperature and then poured into the water and extracted with dichloromethane (DCM). The combined organic extracts were dried over anhydrous MgSO_4_, filtered and concentrated. The crude was purified by FCC on silica gel.

#### 2.3.1. 2-(1-(4-methoxyphenyl)-2-arylvinyl)pyridine **6**

0.05 mL of benzaldehyde was used (0.486 mmol, 1.04 eq). The reaction lasted 25 min. The crude product was purified by FCC using the CyHex/EtOAc: 95/5 and the 2-(1-(4-méthoxyphenyl) -2-phenylvinyl) pyridine was afforded **6** in two separable *E* and *Z* isomers.

*Z* isomer (**6a**): 31.6 mg of a yellow oil in 24% yield. **TLC:** CyHex/EtOAc: 80/20, R*_f_* = 0.36. **IR ν (cm^−1^):** 3061, 3029 (ν_Csp2-H)_; 2922 (ν_Csp3-H_); 2857 (ν_OMe_); 1604, 1582, 1511 and 1462 (ν_C=C_); 1246 (ν_asym C-O-C_); 1031 (ν_sym C-O-C_); 860 (*δ*_Csp2-H p-substitution_). **^1^H NMR (400 MHz, CDCl_3_) *δ* (ppm):** 8.78 (ddd, *J_12–11_* = 4.7 Hz, *J_12–10_* = 1.9 Hz, *J_12–9_* = 0.9 Hz, 1H, H12, 1H, H12), 8.12–8.08 (m, 4H, H15, H16, H18, H19), 7.66–7.62 (m, 1H, H10), 7.48–7.44 (m, 3H, H3, H7, H17), 7.29–7.27 (m, 1H, H11), 7.07(s, 1H, H13), 6.96–6.93 (m, 1H, H9), 6.85 (d, *J_4–3, 6–7_* = 8 Hz, 2H, H4, H6), 3.79 (s, 3H, H20, OMe). **^13^C NMR (100 MHz, CDCl_3_) *δ* (ppm):** 159.4 (C5), 159.1 (C8), 149.7 (C12), 140.9 (C1), 137.1 (C10), 134.4 (C2), 133.6 (C17), 130.1 (C15, C16, C18, C19), 129.8 (C14), 129.4 (C9), 128.8 (C13), 128.5 (C3, C7), 120.0 (C11), 113.9 (C4, C6), 55.3 (C20). **LRMS:** (ES+, CV = 30). *m*/*z* = 575 [2M + H]^+^, 288 [M + H]^+^, 257 [M − OMe]^+^. **HRMS:** calcd. for C_20_H_17_NOH [M + H]^+^ (288.1383); found (288.1383).

*E* isomer (**6b**): 31.8 mg of white amorphous solid 44% yield. **TLC:** CyHex/EtOAc: 80/20, R*_f_* = 0.58. **IR ν (cm^−1^):** 3061, 3029 (ν_Csp2-H)_; 2922 (ν_Csp3-H_); 2857 (ν_OMe_); 1604, 1582, 1511 and 1462 (ν_C=C)_; 1246 (ν_asym C-O-C)_; 1031 (ν_sym C-O-C_). **^1^H NMR (400 MHz, CDCl_3_) *δ* (ppm):** 8.65 (ddd, *J_12–11_* = 4.7 Hz, *J_12–10_* = 1.9 Hz, *J_12–9_* = 0.9 Hz, 1H, H12), 7.76 (s, 1H, H13), 7.55 (td, *J_10–9,10–11_* = 7.8 Hz, *J_10–12_* = 1.8 Hz, 1H, H10), 7.17- 7.09 (m, 8H, H3, H7, H11, H15, H16, H17, H18, H19), 7.02 (d, *J_9–10_* = 7.9 Hz, 1H, H9), 6.95–6.92 (m, 2H, H4, H6), 3.85 (s, 3H, H20, OMe). **^13^C NMR (100 MHz, CDCl_3_) *δ* (ppm):** 159.4 (C5), 159.2 (C8), 149.2 (C12), 140.1 (C1), 137.0 (C13), 136.5 (C10), 131.5 (C3, C7), 131.3 (C2), 130.1 (C15, C19), 129.2 (C14), 128.0 (C16, C18), 127.3 (C17), 122.6 (C9), 122.0 (C11), 114.5 (C4, C6), 55.3 (C20). **LRMS:** (ES+, CV = 30) *m*/*z* = 288 [M + H]^+^, 271 [M − Me]^+^.

#### 2.3.2. 2-(1-(4-methoxyphenyl)-2-(p-tolyl)vinyl)pyridine **7**

0.059 mL of *p-*tolylaldehyde was used (0.486 mmol, 1.04 eq). The reaction lasted 12 min. The crude product was purified by FCC using a gradient system of CyHex/EtOAc (from 1 to 5% of EtOAc), the 2-(1,2-bis(4 methoxyphenyl)vinyl)pyridine **7** was afforded in two separable *E* and *Z* isomers.

*Z* isomer (**7a**): 20 mg of yellow oil in 14% yield. **TLC:** CyHex/EtOAc:80:20, R*_f_* = 0.2. **^1^H NMR (400 MHz, DMSO*-d_6_*) *δ* (ppm):** 8.66 (ddd, *J_12–11_* = 4.7 Hz, *J_12–10_* = 1.9 Hz, *J_12–9_* = 0.9 Hz, 1H, H12), 7.76 (td, *J_10–9, 10–11_* = 7.7 Hz, *J_10–12_* = 1.7 Hz, 1H, H10), 7.36 (ddd, *J_11–10_* = 7.5 Hz, *J_11–12_* = 4.9 Hz, *J_11–9_* = 1.2 Hz, 1H, H11), 7.20–7.12 (m, 3H, H9, H15, H19), 7.06 (s, 1H, H13), 6.95–6.86 (m, 4H, H3, H7, H16, H18), 6.75 (d, *J_4–3, 6–7_* = 7.9 Hz, 2H, H4, H6), 3.76 (s, 3H, H20, OMe), 2.22–2.16 (s, 3H, H21, Me). **^13^C NMR (100 MHz, DMSO*-d_6_*) *δ* (ppm):** 159.12 (C5), 158.78 (C8), 149.95 (C12), 140.09 (C1), 136.96 (C10), 136.21 (C17), 133.98 (C2), 133.89 (C4, C6), 128.87 (C14), 128.63 (C3, C7), 127.99 (C15, C19), 126.81 (C13), 125.18 (C9), 122.55 (C11), 113.77 (C16, C18), 55.15 (C20), 28.68 (C21). **LRMS:** (ES+, CV = 30). *m*/*z* = 324.25 [M + Na]^+^, 303.23 [M + 2H]^+^, 302.24 [M + H]^+^. **HRMS:** calcd. for C_21_H_19_NOH [M + H]^+^ (302.1539); found (302.1540).

*E* isomer (**7b**): 30 mg of yellow amorphous solid in 21% yield. **TLC:** CyHex/EtOAc: 80/20, R*_f_* = 0.4. **IR ν (cm^−1^):** 3000 (ν_Csp2-H_); 2940, 2921 (ν _Csp3-H)_; 2854 (ν_OMe_); 1604, 1580, 1509 and 1462 (ν_C=C_); 1240 (ν_C-O_); 1174 (ν_C-N_); 815 (*δ*_Csp2-H p-substitution_). **^1^H NMR (400 MHz, DMSO-*d_6_*) *δ* (ppm):** 8.60 (ddd, *J_12–11_* = 4.7 Hz, *J_12–10_* = 1.9 Hz, *J_12–9_* = 0.9 Hz, 1H, H12), 7.74 (s, 1H, H13), 7.68 (td, *J_10–11, 10–9_* = 7.7 Hz, *J_10–12_* = 1.9 Hz, 1H, H10), 7.28–7.24 (m, 1H, H11), 7.10 (d, *J_3–4,7–8_* = 8.6 Hz, 2H, H3, H7), 7.02–6.95 (m, 4H, H15, H19, H16, H18), 6.94–6.93 (m, 1H, H9), 3.80 (s, 3H, H20, OMe), 2.08 (s, 3H, H21, Me). **^13^C NMR (100 MHz, DMSO-*d_6_*) *δ* (ppm):** 158.6 (C5), 158.2 (C8), 149.0 (C12), 136.8 (C1), 136.7 (C10), 134.5 (C2), 133.6 (C17), 130.9 (C3, C7), 129.8 (C16, C18), 129.5 (C13), 128.7 (C14), 124.4 (C15, C19), 122.1 (C9), 121.7 (C11), 114.6 (C6, C4), 55.1 (C20), 30.7 (C21). **LRMS:** (ES+, CV = 30). *m*/*z* = 324 [M + Na]^+^, 303.23 [M + 2H]^+^, 302.17 [M + H]^+^, 211.12 [M − CH_3_C_6_H_5_]^+^.

#### 2.3.3. 2-(1,2-bis(4-methoxyphenyl)vinyl)pyridine **8**

0.056 mL of *p*-anisaldehyde was used (0.486 mmol, 1.04 eq). The reaction lasted 3 h. The crude product was purified by FCC using the CyHex/EtOAc 95:5, the 2-(1,2-bis(4-methoxyphenyl)vinyl)pyridine **8** was afforded in two separable *E* and *Z* isomers.

*Z* Isomer (**8a**): 34.4 mg of a yellow colored oil in 24% yield. **TLC:** CyHex/EtOAc: 80/20, R*_f_* = 0.5. **IR ν (cm^−1^):** 2932 (ν _Csp3-H)_; 2855 (ν_OMe_); 1604, 1509 and 1461 (ν_C=C_); 1244 (ν_C-O_); 1176 (ν_C-N_); 829 (*δ*_Csp2-H p-substitution_). **^1^H NMR (400 MHz, DMSO-*d_6_*) *δ* (ppm):** 8.66 (ddd, *J_12–11_* = 4.7 Hz, *J_12–10_* = 1.9 Hz, *J_12–9_* = 0.9 Hz, 1H, H12), 7.78 (td, *J_10–11, 10–9_* = 7.9 Hz, *J_10–12_* = 1.8 Hz, 1H, H10), 7.38–7.35 (m, 1H, H9), 7.17 (m, 3H, H11, H3, H7), 7.03 (s, 1H, H13), 6.89 (d, *J_4–3, 6–7_* = 7.2 Hz, 2H, H4, H6), 6.80 (d, *J_16–15, 18–19_* = 8.6 Hz, 1H, H16, H18), 6.70 (d, *J_15–16, 19–18_* = 7.4 Hz, 2H, H15, H19), 3.74 (s, 3H, H21, Me), 3.67 (s, 3H, H20, OMe). **^13^C NMR (100 MHz, DMSO-*d_6_*) *δ* (ppm):** 159.7 (C17), 159.1 (C5), 158.6 (C8), 150.4 (C12), 139.2 (C1), 137.5 (C10), 134.5 (C2), 130.7 (C16, C18), 129.7 (C14), 128.2 (C3, C7), 126.9 (C13), 125.6 (C11), 123.0 (C9), 114.21 (C15, C19), 113.9 (C6, C4), 55.8 (C20), 54.7 (C21). **LRMS:** (ES+, CV = 30). *m*/*z* = 657.28 [2M + Na]^+^, 635.35 [2M + H]^+^, 318.18 [M + H]^+^, 302.24 [M − Me]^+^. **HRMS:** calcd. for C_21_H_19_NO_2_H [M + H]^+^ (318.1489); found (318.1489).

*E* isomer (**8b**): 92.3 mg of a yellow-colored oil in 60% yield. **TLC:** CyHex/EtOAc: 80/20, R*_f_* = 0.6. **IR ν (cm^−1^):** 2932 (ν _Csp3-H)_; 2855 (ν_OMe_); 1604, 1509 and 1461 (ν_C=C_); 1244 (ν_C-O_); 829 (*δ*_Csp2-H p-substitution_). **^1^H NMR (400 MHz, DMSO-*d_6_*) *δ*** 8.59 (ddd, *J_12–11_* = 4.7 Hz, *J_12–10_* = 1.9 Hz, *J_12–9_* = 0.9 Hz, 1H, H12), 7.76 (s, 1H, H13), 7.66 (td, *J_10–9, 10–11_* = 7.7 Hz, *J_10–12_* = 1.8 Hz, 1H, H10), 7.24–7.21 (m, 1H, H11), 7.11 (d, *J_3–4, 7–6_* = 8.7 Hz, 2H, H3, H7), 7.04–6.98 (m, 4H, H6, H4, H16, H18), 6.91 (d, *J_9–10_* = 8 Hz, 1H, H9), 6.47 (d, *J_15–16, 19–18_* = 8.8 Hz, 2H, H15, H19), 3.81 (s, 3H, H21, Me), 3.63 (s, 3H_,_ H20, OMe). **^13^C NMR (100 MHz, DMSO-*d_6_*) *δ* (ppm):** 159.1 (C17), 159.0 (C5), 158.7 (C8), 149.4 (C12), 138.0 (C1), 137.1 (C10), 134.5 (C2), 131.4 (C3, C7, C16, C18), 129.9 (13), 129.4 (C14), 122.3 (C11), 121.9 (C9), 115.2 (C6, C4), 114.1 (C15, C19), 55.5 (C20, C21). **LRMS:** (ES+, CV = 30). *m*/*z* = 340.19 [M + Na]^+^, 308.18 [M + H]^+^.

#### 2.3.4. 2-(2-(4-isopropylphenyl)-1-(4-methoxyphenyl)vinyl)pyridine **9**

0.059 mL of 4-isopropylbenzaldehyde was used (0.486 mmol, 1.04 eq). The reaction lasted 15 min. The crude product was purified by FCC using a gradient system of CyHex/EtOAc (from 2 to 10% of EtOAc), the 2-(1,2-bis(4 methoxyphenyl)vinyl)pyridine **9** was afforded in two separable *E* and *Z* isomers.

*Z* isomer (**9a**): 100 mg of yellow amorphous solid in 65% yield. **TLC:** CyHex/EtOAc:80/20, R*_f_* = 0.30. **IR ν (cm^−1^):** 3001 (ν_Csp2-H_); 2958, 2927 (ν_Csp3-H)_; 2835 (ν_OMe_); 1602, 1583, 1509 and 1460 (ν_C=C_); 1244 (ν_C-O_); 1179 (ν_C-N_); 826 (*δ*_Csp2-H p-substitution_). **^1^H NMR (400 MHz, DMSO-*d_6_*) *δ* (ppm):** 8.68 (ddd, *J_12–11_* = 4.7 Hz, *J_12–10_* = 1.9 Hz, *J_12–9_* = 0.9 Hz, 1H, H12, 1H, H12), 7.80 (td, *J_10–11, 10–9_* = 7.7 Hz, *J_10–12_* = 1.8 Hz, 1H, H10), 7.40 (ddd, *J_9–10_* = 7.6 Hz, *J_9–11_* = 4.9 Hz, *J_9–12_ =* 1.2 Hz, 1H, H9), 7.21–7.16 (m, 3H, H11, H15, H19), 7.07 (s, 1H, H13), 7.02–6.98 (m, 2H, H3, H7), 6.93–6.88 (m, 2H, H16, H18), 6.78 (d, *J_4–3_* = *J_6–7_* = 8.3 Hz, 2H, H4, H6), 3.76 (s, 3H, H20, OMe), 2.85–2.71 (m, 1H, H21), 1.23 (d, *J_22–21_*= 6.9 Hz, 6H, H22). **^13^C NMR (100 MHz, DMSO-*d_6_*) *δ* (ppm):** 159.56 (C5), 159.25 (C8), 150.45 (C12), 147.67 (C17), 140.41 (C1), 137.55 (C10), 134.74 (C14), 134.33 (C3, C7), 129.41 (C2), 128.41 (C15, C19), 127.17 (C13), 126.44 (C16, C18), 125.60 (C9), 123.10 (C11), 114.26 (C4, C6), 55.62 (C20), 33.48 (C21), 29.46 (C22). **LRMS:** (ES+, CV = 30) *m*/*z*: 681.58 [2M + Na]^+^; 352.25 [M + Na]^+^; 331.31 [M + 2H]^+^; 330.24 [M + H]^+^. **HRMS:** calcd. for C_23_H_23_NOH [M + H]^+^ (308.1281); found (308.1258).

*E* isomer (**9b**): 46 mg of a yellow-colored oil in 30% yield. **TLC:** CyHex/EtOAc: 80/20, R*_f_* = 0.7. **IR ν (cm^−1^):** 3001 (ν_Csp2-H_); 2958, 2927 (ν_Csp3-H)_; 2835 (ν_OMe_); 1602, 1583, 1509 and 1460 (ν_C=C_); 1244 (ν_C-O_); 1179 (ν_C-N_); 826 (*δ*_Csp2-H p-substitution_). **^1^H NMR (400 MHz, DMSO-*d_6_*) *δ* (ppm):** 8.60 (ddd, *J_12–11_* = 4.7 Hz, *J_12–10_* = 1.9 Hz, *J_12–9_* = 0.9 Hz, 1H, H12, 1H, H12), 7.78 (s, 1H, H13), 7.63 (td, *J_10–11, 10–9_* = 7.7 Hz, *J_10–12_* = 1.9 Hz, 1H, H10), 7.27–7.23 (m, 1H, H11), 7.12 (d, *J_3–4,7–8_* = 8.7 Hz, 2H, H3, H7), 7.06–7.02 (m, 4H, H15, H19, H16, H18), 6.95 (d, *J_4–3, 6–7_* = 8.2 Hz, 2H, H4, H6), 6.94–6.91 (m, 1H, H9), 3.81 (s, 3H, H20, OMe), 2.82–2.80 (m, 1H, H21), 1.13 (d, *J_22–21_* = 6.8 Hz, 6H, H22). **^13^C NMR (100 MHz, DMSO-*d_6_*) *δ* (ppm):** 158.3 (C5), 158.1 (C8), 150.6 (C12), 145.5 (C17), 139.0 (C1), 136.5 (C10), 135.7 (C2), 134.8 (C3, C7), 129.6 (C14), 129.1 (C15, C19), 127.2 (C13), 127.0 (C16, C18), 125.6 (C9), 123.1 (C11), 113.8 (C4, C6), 55.5 (C20), 33.8 (C21), 29.5 (C22). **LRMS:** (ES+, CV = 30) *m*/*z*: 681.58 [2M + Na]^+^; 352.25 [M + Na]^+^; 331.23 [M + 2H]^+^; 330.24 [M + H]^+^.

#### 2.3.5. 2-(2-(4-(tert-butyl)phenyl)-1-(4-methoxyphenyl)vinyl)pyridine **10**

0.084 mL of 4-*tert-*butylbenzaldehyde was used (0.486 mmol, 1.04 eq). The reaction lasted 20 min. The crude product was purified by FCC using a gradient system of CyHex/EtOAc (from 2 to 10% of EtOAc), the 2-(1,2-bis(4 methoxyphenyl)vinyl)pyridine **10** was afforded in two separable *E* and *Z* isomers.

*Z* isomer (**10a**): 89 mg of a white amorphous solid in 56%. **TLC:** CyHex/EtOAc:80/20, R*_f_* = 0.2. **IR ν (cm^−1^):** 3047, 3005 (ν_Csp2-H_); 2956, 2931 (ν _Csp3-H)_; 2866 (ν_OMe_); 1603, 1593, 1509 and 1462 (ν_C=C_); 1244 (ν_C-O_); 1181 (ν_C-N_); 827 (*δ*_Csp2-H p-substitution_). **^1^H NMR (400 MHz, DMSO*-d_6_*) *δ* (ppm):** 8.68 (ddd, *J_12–11_* = 4.7 Hz, *J_12–10_* = 1.9 Hz, *J_12–9_* = 0.9 Hz, 1H, H12), 7.80 (td, *J_10–11, 10–9_* = 7.7 Hz, *J_10–12_* = 1.8 Hz, 1H, H10), 7.40 (m, 1H, H11), 7.20–7.14 (m, 5H, H9, H3, H7, H15, H19), 7.06 (s, 1H, H13), 6.92–6.89 (m, 2H, H6, H4), 6.79 (d, *J_16–17, 18–19_* = 8.4 Hz, 2H, H16, H18), 3.75 (s, 3H, H20, OMe), 1.20 (s, 9H, H22). **^13^C NMR (100 MHz, DMSO*-d_6_*) *δ* (ppm):** 158.91 (C5), 158.57 (C8), 149.80 (C17), 149.21(C12), 139.76 (C1), 136.87 (C10), 133.67 (C2), 129.14 (C14), 128.48 (C3, C7), 127.72 (C16, C18), 126.94 (C13), 126.34 (C15, C19), 124.90 (C9), 123.89 (C11), 122.43, 113.58 (C4, C6), 54.96 (C20), 34.00 (C21), 30.76 (C22). **LRMS:** (ES+, CV = 30) *m*/*z*: 709.37 [2M + Na]^+^; 687.42 [2M + H]^+^; 366.22 [M + Na]^+^; 345.27 [M + 2H]^+^; 344.28 [M + H]^+^. **HRMS:** calcd. for C_24_H_25_NOH [M + H]^+^ (344.2009); found (344.2009).

*E isomer* (**10b**): 59 mg of a yellow amorphous solid 37%. **TLC:** CyHex/EtOAc:80/20, R*_f_* = 0.5. **IR ν (cm^−1^):** 3050, 3005 (ν_Csp2-H_); 2957, 2926 (ν_Csp3-H)_; 2858 (ν_OMe_); 1602, 1581, 1509 and 1461 (ν_C=C_); 1240 (ν_C-O_); 1178 (ν_C-N_); 832 (*δ*_Csp2-H p-substitution_). **^1^H NMR (400 MHz, DMSO*-d_6_*) *δ* (ppm):** 8.61 (ddd, *J_12–11_* = 4.7 Hz, *J_12–10_* = 1.9 Hz, *J_12–9_* = 0.9 Hz, 1H, H12, 1H, H12), 7.80 (s, 1H, H13), 7.68 (td, *J*_10–11, 10–9_ = 7.7 Hz, *J_10–12_* = 1.9 Hz, 1H, H10), 7.26 (m, 1H, H11), 7.20 (d, *J_15–16, 19–18_* = 8.5 Hz, 2H, H15, H19), 7.15–7.11 (m, 2H, H3, H17), 7.07–7.03 (m, 2H, H4, H6), 6.99 (d, *J_16–17, 18–19_* = 8.5 Hz, 2H, H16, H18), 6.92 (d, *J_9–10_* = 8.0 Hz, 1H, H9), 3.82 (s, 3H, H20, OMe), 1.21 (s, 9H, H22). **^13^C NMR (100 MHz, DMSO*-d_6_*) *δ* (ppm):** 158.68 (C5), 158.12 (C8), 150.09 (C17), 149.07 (C12), 138.92 (C1), 136.72 (C10), 133.54 (C2), 131.72 (C3, C7), 130.84 (C14), 129.55 (C13), 129.41 (C16, C18), 124.99 (C15, C19), 122.12 (C9), 121.65 (C11), 114.75 (C6, C4), 55.09 (C20), 34.30 (C21), 30.95 (C22). **LRMS:** (ES+, CV = 30) *m*/*z*: 709.37 [2M + Na]^+^; 366.22 [M + Na]^+^; 345.27 [M + 2H]^+^; 344.28 [M + H]^+^, 214.16 [M − (CH_3_)_3_CHC_6_H_4_]^+^.

#### 2.3.6. 4-(2-(4-methoxyphenyl)-2-(pyridin-2-yl)vinyl)phenol **11b**

61.2 mg of 4-hydroxybenzaldehyde was used (0.486 mmol, 1.04 eq). The reaction lasted 10 min. The crude product was purified by FCC using a gradient system of CyHex/EtOAc (from 1 to 10% of EtOAc), the 4-(2-(4-methoxyphenyl)-2-(pyridin-2-yl)vinyl)phenol **11b** was afforded in two *E* and *Z* isomers but only *E* isomer was isolated.

*E* isomer (**11b**): 48 mg of a yellow amorphous solid in 34% yield. **TLC:** CyHex/EtOAc: 80/20, R*_f_* = 0.24. **IR ν (cm^−1^):** 3512 (ν_OH_); 3062, 3002 (ν_Csp2-H_); 2957, 2920 (ν_Csp3-H)_; 2853 (ν_OMe_); 1602, 1586, 1507 and 1462 (ν_C=C_); 1242 (ν_C-O_); 1171 (ν_C-N_); 831 (*δ*_Csp2-H p-substitution_). **^1^H NMR (400 MHz, DMSO-*d*_6_) *δ* (ppm):** 9.58 (s, 1H, OH), 8.59 (ddd, *J_12–11_* = 4.7 Hz, *J_12–10_* = 1.9 Hz, *J_12–9_* = 0.9 Hz, 1H, H12), 7.72 (s, 1H, H13), 7.70–7.62 (m, 1H, H10), 7.24–7.19 (m, 1H, H11), 7.12–7.08 (m, 2H, H15, H19), 7.06–7.01 (m, 2H, H, H16, H18 ), 6.92–6.87 (m, 3H, H9, H3, H7), 6.61–6.50 (m, 2H, H4, H6), 3.85–3.78 (s, 3H, H20, OMe). **^13^C NMR (100 MHz, DMSO*-d_6_*) *δ* (ppm):** 158.57 (C5), 158.48 (C17), 157.03 (C8), 149.00 (C12), 136.63 (C1), 131.5 (C2), 131.12 (C3, C7, C15, C19), 129.90 (C13), 127.40 (C14), 121.75 (C9), 121.32 (C11), 115.05 (C16, C18), 114.72 (C4, C6), 113.84 (C14), 55.09 (C20). **LRMS:** (ES+, CV = 30) *m*/*z*: 629.26 [2M + Na]^+^; 326.15 [M + Na]^+^; 3034.21 [M + H]^+^; 200.20 [M − C_6_H_6_OHCH]^+^. **HRMS:** calcd. for C_20_H_17_NO_2_H [M + H]^+^ (304.1332); found (304.1333).

#### 2.3.7. 2-(1-(4-methoxyphenyl)-2-(4-(trifluoromethyl)phenyl)vinyl)pyridine **12**

0.064 mL of 4-(trifluoromethyl) benzaldehyde was used (0.486 mmol, 1.04 eq). The reaction lasted 15 min. The crude product was purified by FCC using a gradient system of CyHex/EtOAc (from 1 to 10% of EtOAc), the 2-(1-(4-methoxyphenyl)-2-(4-(trifluoromethyl)phenyl) vinyl)pyridine **12** was afforded in two separable *E* and *Z* isomers.

*Z* isomer (**12a**): 60 mg of yellow oil in 36% yield. **TLC:** CyHex/EtOAc: 80/20, R*_f_* = 0.25. **IR ν (cm^−1^):** 3020, 3013 (ν_Csp2-H_); 2921 (ν _Csp3-H)_; 2846 (ν_OMe_); 1605, 1579, 1510 and 1464 (ν_C=C_); 1244 (ν_C-O_); 1111 (ν_C-N_); 834 (*δ*_Csp2-H p-substitution_). **^1^H NMR (400 MHz, DMSO-*d*_6_) *δ* (ppm):** 8.67 (ddd, *J_12–11_* = 4.7 Hz, *J_12–10_* = 1.9 Hz, *J_12–9_* = 0.9 Hz, 1H, H12), 7.79 (td, *J*_10–11_= 7.7, 1.8 Hz, 1H, H10), 7.51 (d, *J_15–16, 19–18_* = 8.2 Hz, 2H, H15, H19), 7.42–7.37 (m, 1H, H11), 7.27–7.15 (m, 4H, H13, H9, H3, H7), 7.05 (d, *J_16–15, 18–19_* = 8.1 Hz, 2H, H16, H18), 6.92 (d, *J_4–3, 6–7_* = 8.9 Hz, 2H, H4, H6), 3.82–3.66 (s, 3H, H20, OMe). **^13^C NMR (100 MHz, DMSO-*d*_6_) *δ* (ppm):** 158.96 (C5), 157.64 (C8), 149.25 (C12), 142.50 (C1), 140.84 (C10), 136.88 (C2), 131.9 (C17), 130.95 (C16, C19, C3, C7), 129.89 (C14), 128.22 (C14), 126.00 (C13), 124.93 (C16, C18), 124.89 (C21), 122.29 (C9), 121.51 (C11), 114.72 (C4, C6), 55.11 (C20). **LRMS:** (ES+, CV = 30) *m*/*z*: 733.29 [2M + Na]^+^; 378.13 [M + Na]^+^; 357.19 [M + 2H]^+^; 307.17 [M + 2H]^+^; 356.16 [M + H]^+^; 288.89 [M − CF_3_]^+^. **HRMS:** calcd. for C_21_H_16_F_3_NOH [M + H]^+^ (356.1256); found (396.1257).

*E* isomer (**12b**): 76 mg of a white amorphous solid in 46% yield. **TLC:** CyHex/EtOAc: 80/20, R*_f_* = 0.44. **IR ν (cm^−1^):** 3043, 3005 (ν_Csp2-H_); 2920 (ν _Csp3-H)_; 2846 (ν_OMe_); 1605, 1579, 1509 and 1464 (ν_C=C_); 1244 (ν_C-O_); 1111 ((ν_C-N_); 831 (*δ*_Csp2-H p-substitution_). **^1^H NMR (400 MHz, DMSO-*d*_6_) *δ* (ppm):** 8.63 (ddd, *J_12–11_* = 4.7 Hz, *J_12–10_* = 1.9 Hz, *J_12–9_* = 0.9 Hz, 1H, H12), 7.82 (s, 1H, H13), 7.73 (td, *J_10–11, 10–9_* = 7.7 Hz, *J_10–12_* = 1.9 Hz, 1H, H10), 7.54 (d, *J_15–16_ = J_19–18_* = 8.3 Hz, 2H, H15, H19), 7.33 (ddd, *J_11–10_* = 7.5 Hz, *J_11–12_* = 4.7, *J_11–9_* = 1.1 Hz, 1H, H11), 7.29–7.25 (m, 2H, H16, H18), 7.13 (d, *J_3–4_* = *J_7–6_* = 8.7 Hz, 2H, H3, H7), 7.07–6.99 (m, 3H, H9, H4, H6), 3.80–3.79 (s, 3H, H20, OMe). **^13^C NMR (100 MHz, DMSO-*d*_6_) *δ* (ppm):** 158.96 (C5), 157.64 (C8), 149.25 (C12), 142.50 (C1), 140.84 (C10), 136.88 (C2), 131.57 (C17), 130.95 (C3, C7), 130.01 (C16, C18), 129.89 (C21), 129.20 (C14), 128.22 (C13), 124.97 (C15, C19), 122.82 (C9), 122.29 (C11), 114.72 (C4, C6), 55.11 (C20). **LRMS:** (ES+, CV = 30) *m*/*z*: 378.07 [M + Na]^+^; 357.12 [M + 2H]^+^; 365.05 [M + H]^+^; 212.15 [M − CF_3_C_6_H_5_]^+^.

#### 2.3.8. 2-(2-(4-bromophenyl)-1-(4-methoxyphenyl)vinyl)pyridine **13**

76.8 mg of 4-bromobenzaldehyde was used (0.486 mmol, 1.04 eq). The reaction lasted 15 min. The crude product was purified by FCC using a gradient system of CyHex/EtOAc (90/10), the 2-(2-(4-bromophenyl)-1-(4-methoxyphenyl)vinyl)pyridine **13** was afforded in two separable *E* and *Z* isomers.

*Z* isomer (**13a**): 57.5 mg of a yellow oil 34% yield. **TLC:** CyHex/EtOAc: 80/20, R*_f_* = 0.34. **IR ν (cm^−1^):** 3048, 3000 (ν_Csp2-H_); 2950, 2927 (ν_Csp3-H)_; 2837 (ν_OMe_); 1601, 1582, 1509 and 1473 (ν_C=C_); 1245 (ν_C-O_); 1179 (ν_C-N_); 818 (*δ*_Csp2-H p-substitution_). **^1^H NMR (400 MHz, DMSO-*d_6_* ) *δ* (ppm):** 8.65 (ddd, *J_12–11_* = 4.7 Hz, *J_12–10_* = 1.9 Hz, *J_12–9_* = 0.9 Hz, 1H, H12), 7.78 (t, *J_10–11, 10–9_* = 7.7 Hz, 1H, H10), 7.40–7.31 (m, 1H, H11), 7.32 (d, *J_15–16, 19–18_* = 8.5 Hz, H15, H19), 7.19 (d, *J_3–4, 7–6_* = 8.8 Hz, 2H, H3, H7), 7.16 (d, *J_9–10_* = 7.8 Hz, 1H, H9), 7.10 (s, 1H, H13), 6.91 (d, *J_4–5, 6–7_* = 8.8 Hz, 4H, H4, H6), 6.81 (d, *J_16–15, 18–19_* = 8.5 Hz, 2H, 2H, H16, H18), 3.75 (s, 3H, H20, OMe). **^13^C NMR (100 MHz, DMSO-*d_6_*) *δ* (ppm):** 159.0 (C5), 158.5 (C8), 150.0 (C12), 141.7 (C1), 137.1 (C10), 136.2 (C2), 133.4 (14), 130.9 (C15, C19), 130.8 (C16, C18), 128.2 (C3, C7), 125.6 (C13), 125.1 (C9), 122.8 (C11), 119.9 (C17), 113.8 (C4, C6), 55.1 (C20). **LRMS:** (ES+, CV = 30) *m*/*z*: 754.92 [2M + Na]^+^; 366.92 [M]^+^; 278.98 [M − Br]^+^. **HRMS:** calcd. for C_20_H_16_BrNOH [M + H]^+^ (366.0488); found (366.0490).

*E* isomer (**13b**): 84.7 mg of a white amorphous solid product in 50% yield. **TLC:** CyHex/EtOAc: 80/20, R*_f_* = 0.53. **IR ν (cm^−1^):** 3065, 3001 (ν_Csp2-H_); 2957, 2931 (ν _Csp3-H)_; 2838 (ν_OMe_); 1602, 1576, 1507 and 1461 (ν_C=C_); 1240 (ν_C-O_); 1173 (ν_C-N_); 818 (*δ*_Csp2-H p-substitution_). **^1^H NMR (400 MHz, DMSO-*d_6_*) *δ* (ppm):** 8.61 (ddd, *J_12–11_* = 4.7 Hz, *J_12–10_* = 1.9 Hz, *J_12–9_* = 0.9 Hz, 1H, H12), 7.73–7.69 (m, 2H, H10, H13), 7.29 (d, *J_15–16, 19–18_* = 8.5 Hz, 1H, H15, H19), 7.29–7.27 (m, 1H, H11), 7.10 (d, *J_3–4, 7–6_* = 8.7 Hz, 2H, H3, H7), 7.02–6.98 (m, 5H, H9, H4, H6, H16, H18), 6.95–6.92 (m, 2H, H4, H6), 3.80 (s, 3H, H20, OMe). **^13^C NMR (100 MHz, DMSO-*d_6_*) *δ* (ppm):** 158.8 (C5), 157.8 (C8), 149.2 (C12), 140.8 (C1), 136.8 (C10), 135.8 (C2), 131.4 (C15,C19), 131.1 (C16, C18), 130.9 (C3, C7), 130.1 (C14), 128.5 (C13), 125.1 (C9), 122.5 (C11), 122.0 (C17), 114.7 (C4, C6), 55.1 (C20). **LRMS:** (ES+, CV = 30) *m*/*z*: 369.12 [M+3H]^+^; 367.93 [M + H]^+^.

#### 2.3.9. 2-(2-(4-chlorophenyl)-1-(4-methoxyphenyl)vinyl)pyridine **14**

70.4 mg of 4-chlorobenzaldehyde was used (0.486 mmol, 1.04 eq). The reaction lasted 10 min. The crude was purified by FCC using a gradient system of CyHex/EtOAc (from 1 to 5% of EtOAc). The 2-(2-(4-bromophenyl)-1-(4-methoxyphenyl)vinyl)pyridine **14** was afforded in two separable *E* and *Z* isomers.

*Z* isomer (**14a**) 17.8 mg of an orange-colored oil 12% yield. **TLC:** CyHex/EtOAc: 80/20, R*_f_* = 0.37. **IR ν (cm^−1^):** 3080, 3011 (ν_Csp2-H_); 2940, 2926 (ν_Csp3-H)_; 2838 (ν_OMe_); 1604, 1578, 1508 and 1461 (ν_C=C_); 1241 (ν_C-O_); 1174 (ν_C-N_); 820 (*δ*_Csp2-H p-substitution_). **^1^H NMR (400 MHz, DMSO*-d_6_* ) *δ* (ppm):** 8.65 (ddd, *J_12–11_* = 4.7 Hz, *J_12–10_* = 1.9 Hz, *J_12–9_* = 0.9 Hz, 1H, H12), 7.78 (td, *J_10–9, 10–11_* = 7.7 Hz, *J_10–12_* = 1.9 Hz, 1H, H10), 7.40–7.37 (m, 1H, H11), 7.20- 7.15 (m, 5H, H3, H7, H9, H15, H19), 7.10 (s, 1H, H13), 6.92–6.86 (m, 4H, H4, H6, H16, H18), 3.75 (s, 3H, H20, OMe). **^13^C NMR (100 MHz, DMSO-*d_6_*) *δ* (ppm):** 159.0 (C5), 158.5 (C8), 150.0 (C12), 141.6 (C1), 137.1 (C10), 135.8 (C2), 133.4 (C14), 131.2 (C17), 130.5 (C16, C18), 128.2 (C3, C7), 128.0 (C15, C19), 125.5 (C13), 125.0 (C9), 122.8 (C11), 113.8 (C6, C4), 55.2 (C20). **LRMS:** (ES+, CV = 30). *m*/*z* = 643.08 [2M]^+^; 344.01 [M + Na]^+^; 322.05 [M + H]^+^; 197.09 [M − C_6_H_4_CHCl]^+^. **HRMS:** calcd. for C_20_H_16_ClNOH [M + H]^+^ (322.0992); found (322.0995).

*E* isomer (**14 b**): 35 mg of a white amorphous solid 24% yield. **TLC:** CyHex/EtOAc: 80/20, R*_f_* = 0.5. **IR ν (cm^−1^):** 3080, 3013 (ν_Csp2-H_); 2940, 2926 (ν_Csp3-H)_; 2838 (ν_OMe_); 1604, 1578, 1508 and 1461 (ν_C=C_); 1241 (ν_C-O_); 1174 (ν_C-N_); 820 (*δ*_Csp2-H p-substitution_). **^1^H NMR (400 MHz, DMSO*-d_6_* ) *δ* (ppm):** 8.62 (ddd, *J_12–11_* = 4.7 Hz, *J_12–10_* = 1.9 Hz, *J_12–9_* = 0.9 Hz, 1H, H12), 7.74 (s, 1H, H13), 7.61 (t, *J_10–9,10–11_* = 7.7 Hz, 1H, H10), 7.30- 7.27 (m, 1H, H11), 7.24 (d, *J_15–16, 19–18_* = 8.6 Hz, 2H, H15, H19), 7.11–7.00 (m, 6H, H3, H7, H16, H18, H4, H6), 6.98 (d, *J_9–10_* = 6.9 Hz, 1H, H9), 3.80 (s, 3H, H20, OMe). **^13^C NMR (100 MHz, DMSO-*d_6_*) *δ* (ppm):** 158.8 (C5), 157.8 (C8), 149.1 (C12), 140.7 (C1), 136.8 (C10), 135.5 (C2), 131.7 (C14), 131.1 (C3, C7), 130.9 (C16,C18), 130.1 (C17), 128.4 (C13), 128.1 (C15,C19), 122.5 (C9), 121.9 (C11), 114.7 (C6, C4), 55.1 (C20). **LRMS:** (ES+, CV = 30). *m*/*z* = 344.06 [M + Na]^+^, 324.12 [M+3H]^+^, 322.12 [M + H]^+^.

#### 2.3.10. 2-(2-(4-fluorophenyl)-1-(4-methoxyphenyl)vinyl)pyridine **15**

0.053 mL of 4-fluorobenzaldehyde was used (0.486 mmol, 1.04 eq). The reaction lasted 15 min. The crude product was purified by FCC using a gradient system of CyHex/EtOAc (from 1 to 5% of EtOAc), the 2-(2-(4-bromophenyl)-1-(4-methoxyphenyl)vinyl)pyridine **15** was afforded in two separable *E* and *Z* isomers.

*Z* isomer (**15a**): 71 mg of yellow amorphous solid in 50% yield. **TLC:** CyHex/EtOAc:80:20, R*_f_* = 0.34. **^1^H NMR (400 MHz, DMSO*-d_6_*) *δ* (ppm):** 8.66 (ddd, *J_12–11_* = 4.7 Hz, *J_12–10_* = 1.8 Hz, *J_12–9_* = 0.9 Hz, 1H, H12), 7.79 (td, *J_10–11, 10–9_* = 7.7 Hz, *J_10–12_* = 1.8 Hz, 1H, H10), 7.39 (ddd, *J_11–10_* = 7.6 Hz, *J_11–12_* = 4.9 Hz, *J_11–9_* = 1.2 Hz, 1H, H11), 7.24–7.15 (m, 3H, H9, H15, H19), 7.13 (s, 1H, H13), 7.03–6.95 (m, 2H, H3, H7), 6.95–6.88 (m, 4H, H4, H6, H16, H18), 3.75 (s, 3H, H20, OMe). **^13^C NMR (100 MHz, DMSO-*d_6_*) *δ* (ppm):** 159.2 (C5), 158.5 (C8), 149.9 (C12), 141.4 (C1), 137.8 (C10), 135.8 (C2), 133.5 (C14), 131.6 (C17), 130.5 (C16, C18), 128.2 (C3, C7), 128.0 (C15, C19), 125.5 (C13), 125.0 (C9), 122.8 (C11), 113.8 (C6, C4), 55.2 (C20). **LRMS:** (ES+, CV = 30) *m*/*z*: 633.29 [2M + Na]^+^; 328.20 [M + Na]^+^; 306.19 [M + H]^+^. **HRMS:** calcd. For C_20_H_16_FNOH [M + H]^+^ (306.1288); found (306.1289).

*E* isomer (**15b**): 43.7 mg of yellow oil in 31% yield. **TLC:** CyHex/EtOAc: 80/20, R*_f_* = 0.5. **IR ν (cm^−1^):** 3056 (ν_Csp2-H_); 2926 (ν_Csp3-H)_; 2857 (ν_Ome_); 1602, 1508 and 1463 (ν_C=C_); 1224 (ν_C-O_); 1156 (ν_C-N_); 832 (*δ*_Csp2-H p-substitution_). **^1^H NMR (400 MHz, DMSO*-d6*) *δ* (ppm):** 8.61 (ddd, *J_12–11_* = 4.7 Hz, *J_12–10_* = 1.9 Hz, *J_12–9_* = 0.9 Hz, 1H, H12), 7.80 (s, 1H, H13), 7.7 (td, *J_10–9, 10–11_* = 7.7 Hz, *J_10–12_* = 1.9 Hz, 1H, H10), 7.34–7.25 (m, 1H, H11), 7.17–7.08 (m, 5H, H,9 H15, H17, H3, H7), 7.07–6.96 (m, 4H, H16, H18, H4, H6), 3.80 (s, 3H, H20, OMe). **^13^C NMR (100 MHz, DMSO-*d_6_*) *δ* (ppm):** 158.9 (C5), 157.8 (C8), 149.5 (C12), 140.7 (C1), 136.8 (C10), 135.7 (C2), 131.7 (C14), 131.1 (C3, C7), 130.4 (C16,C18), 130.3 (C17), 128.4 (C13), 128.7 (C15, C19), 122.5 (C9), 122.1 (C11), 114.7 (C6, C4), 55.5 (C20). **LRMS:** (ES+, CV = 30) *m*/*z*: 633.29 [2M + Na]^+^; 611.27 [2M + H]^+^; 328.20 [M + Na]^+^; 307.17 [M + 2H]^+^; 306.19 [M + H]^+^.

### 2.4. General Procedure for the Synthesis of aryloxyDAM

To a suspension of molecular sieves (480 mg) in anhydrous dichloromethane (4.8 mL) under argon was added carbinol (0.46 mmol, 100 mg), arylboronic acid (1.38 mmol, 3 eq), Cu(OAc)_2_ (0.46 mmol, 84.5 mg, 1 eq) and anhydrous pyridine (0.92 mmol, 0.074 mL, 2 eq). The reaction mixture was refluxed during 24 h at 40 °C under argon. After reaction completion, the mixture was cooled at room temperature, filtered under celite. To recover the product, the celite was washed using dichloromethane and ethyl acetate. The filtrate was concentrated under vacuum and the crude product was purified by FCC on silica gel (CyHex/EtOAc: 80/20).

#### 2.4.1. 2-((4-methoxyphenyl)(phenoxy)methyl)pyridine **17**

168.2 mg of phenylboronic acid (1.38 mmol, 3 eq) was used. After FCC, the 2-((4-methoxyphenyl)(phenoxy)methyl)pyridine **17** was afforded as a yellow oil (94 mg) in 70% yield. **TLC:** CyHex/EtOAc: 60/40, R*_f_* = 0.7. **IR ν (cm^−1^):** 3060, 3007 (ν_Csp2-H_); 2932, 2927 (ν_Csp3-H_); 2837 (ν_OMe_); 1586, 1510 and 1491 and 1470 (ν_C=C_); 1223 (ν_asym C-O-C_); 1030 (ν_sym C-O-C_); 749 (*δ*_Csp2-H p-substitution_). **^1^H NMR (400 MHz, DMSO*-d_6_* ) *δ* (ppm):** 8.51 (ddd, *J_12–11_* = 4.7 Hz, *J_12–10_* = 1.9 Hz, *J_12–9_* = 0.9 Hz, 1H, H12), 7.79 (td, *J_10–9,10–11_* = 7.6 Hz, *J_10–12_* = 1.8 Hz, 1H, H10), 7.58 (d, *J_9–10_* = 8 Hz, 1H, H9), 7.42 (d, *J_3–4,7–6_* = 8.7 Hz, 2H, H3, H7), 7.28–7.20 (m, 3H, H11, H15, H17), 6.98 (d, *J_14–15, 18–17_* = 7.7 Hz, 2H, H14, H18), 6.90–6.86 (m, 3H, H4, H6, H16), 6.39 (s, 1H, H1), 3.70 (s, 3H, H19, OMe). **^13^C NMR (100 MHz, DMSO-*d_6_*) *δ* (ppm):** 160.4 (C13), 158.8 (C5), 157.2 (C8), 149 (C12), 137.2 (C10), 1321 (C2), 129.4 (C15, C17), 128.3 (C3, C7), 122.7 (C9), 120.9 (C16), 120.4 (C11), 115.8 (C14, C18), 113.8 (C4, C6), 80.9 (C1), 55 (C19). **LRMS:** (ES+, CV = 30) *m*/*z*: 314.04 [M + Na]^+^; 292.01 [M + H]^+^; 198.14 [M − C_6_H_5_O]^+^. **HRMS:** calcd. for C_19_H_17_NO_2_H [M + H]^+^ (292.1332); found (292.1752).

#### 2.4.2. 2-((4-methoxyphenyl)(p-tolyloxy)methyl)pyridine **18**

187.6 mg of 4-methylphenylboronic acid (1.38 mmol, 3 eq) was used. After FCC, the 2-((4-methoxyphenyl)(*p*-tolyloxy)methyl)pyridine **18** was afforded as a white solid (122.2 mg) in 87% yield. **TLC:** CyHex/EtOAc: 60/40, R*_f_* = 0.7. **IR ν (cm^−1^):** 3065 (ν_Csp2-H_); 2994, 2917 (ν_Csp3-H_); 2853 (ν_OMe_); 1610, 1588, 1510 and 1473 (ν_C=C_); 1227 (ν_asym C-O-C_); 1032 (ν_sym C-O-C_); 810 (*δ*_Csp2-H p-substitution_). **^1^H NMR (400 MHz, DMSO*-d_6_* ) *δ* (ppm):** 8.62 (ddd, *J_12–11_* = 4.7 Hz, *J_12–10_* = 1.9 Hz, *J_12–9_* = 0.9 Hz, 1H, H12), 7.78 (td, *J_10–9,10–11_* = 7.6 Hz, *J_10–12_* = 1.7 Hz, 1H, H10), 7.57 (d, *J_9–10_* = 8 Hz, 1H, H9), 7.42 (d, *J_3–4,7–6_* = 8.6 Hz, 2H, H3, H7), 7.27–7.24 (m, 1H, H11), 7.01 (d, *J_15–16, 17–18_* = 8.7 Hz, 2H, H15, H17), 6.98–6.85 (m, 4H, H4, H6, H14, H18), 6.41 (s, 1H, H1), 3.70 (s, 3H, H19, OMe), 2.17 (s, 3H, H20, Me). **^13^C NMR (100 MHz, DMSO-*d_6_*) *δ* (ppm):** 160.5 (C5), 158.7 (C8), 155 (C13), 149 (C12), 137.1 (C10), 132.2 (C16), 129.7 (C15, C17), 129.6 (C2), 128.2 (C3, C7), 122.7 (C9), 120.4 (C11), 115.7 (C4, C6), 113.8 (C14, C18), 81 (C1), 55 (C19), 20 (C20). **LRMS** (ES+, CV = 30) *m*/*z*: 328.16 [M + Na]^+^; 306.18 [M + H]^+^; 214.14 [M − C_7_H_7_]^+^; 198.14 [M − C_7_H_7_O]^+^. **HRMS:** calcd. for C_20_H_19_NO_2_H [M + H]^+^ (306.1489); found (306.1288).

#### 2.4.3. 2-((4-methoxyphenoxy)(4-methoxyphenyl)methyl)pyridine **19**

210 mg of 4-methoxyphenylboronic (1.38 mmol, 3 eq) was used. After a FCC, the 2-((4-methoxyphenoxy)(4-methoxyphenyl)methyl)pyridine **19** as a transparent oil (110 mg) in 75% yield. **TLC:** CyHex/EtOAc: 60/40, R*_f_* = 0.7. **IR ν (cm^−1^):** 3065 (ν_Csp2-H_); 2990, 2930 (ν_Csp3-H_); 2837 (ν_OMe_); 1605, 1588, 1507 and 1463 (ν_C=C_); 1216 (ν_asym C-O-C_); 1031 (ν_sym C-O-C_); 823 (*δ*_Csp2-H p-substitution_). **^1^H NMR (400 MHz, DMSO*-d_6_* ) *δ* (ppm):** 8.50 (ddd, *J_12–11_* = 4.7 Hz, *J_12–10_* = 1.9 Hz, *J_12–9_* = 0.9 Hz, 1H, H12), 7.78 (td, *J_10–9,10–11_* = 7.7 Hz, *J_10–12_* = 1.7 Hz, 1H, H10), 7.58 (d, *J_9–10_* = 7.8 Hz, 1H, H9), 7.40 (d, *J_3–4,7–6_* = 8.7 Hz, 2H, H3, H7), 7.27–7.24 (m, 1H, H11), 6.93–6.87 (m, 4H, H4, H6, H15, H17), 6.68 (d, *J_14–15,18–17_* = 9.1 Hz, 2H, H14, H18), 6.28 (s, 1H, H1), 3.70 (s, 3H, H19, OMe), 3.64 (s, 3H, H20, OMe). **^13^C NMR (100 MHz, DMSO-*d_6_*) *δ* (ppm):** 160.6 (C5), 158.7 (C8), 153.5 (C13), 151.1 (C16), 149 (C12), 137.1 (C10), 132.3 (C2), 128.3 (C3, C7), 122.6 (C9), 120.4 (C11), 116.9 (C4, C6), 114.5 (C14, C18), 113.8 (C15, C17), 81.6 (C1), 55.2 (C20). 55 (C19). **LRMS:** (ES+, CV = 30). *m*/*z*: 323.24 [M + 2H]^+^; 322.16 [M + H]^+^; 214.14 [M − C_7_H_7_O]^+^; 198.15 [M − C_7_H_7_O_2_]^+^. **HRMS:** calcd. for C_20_H_19_NO_3_H [M + H]^+^ (322.1438); found (322.1439).

#### 2.4.4. 2-((4-(tert-butyl)phenoxy)(4-methoxyphenyl)methyl)pyridine **20**

245 mg of 4-*tert*butylmethoxyphenylboronic acid (1.38 mmol, 3 eq) was used. After FCC, the 2-((4-(tert-butyl)phenoxy)(4-methoxyphenyl)methyl)pyridine **20** was afforded as a white solid (123.5 mg) in 77% yield. **TLC:** CyHex/EtOAc 60:40, R*_f_* = 0.7. **IR ν (cm^−1^):** 2958, 2927, 2927 (ν_Csp3-H_); 2847 (ν_OMe_); 1609, 1585, 1512 and 1463 (ν_C=C_); 1238 (ν_asym C-O-C_); 1024 (ν_sym C-O-C_); 822 (*δ*_Csp2-H p-substitution_). **^1^H NMR (400 MHz, DMSO*-d_6_* ) *δ* (ppm):** 8.51 (ddd, *J_12–11_* = 4.7 Hz, *J_12–10_* = 1.9 Hz, *J_12–9_*= 0.9 Hz, 1H, H12), 7.79 (td, *J_10–9,10–11_* = 7.8 Hz, *J_10–12_* = 1.8 Hz, 1H, H10), 7.58 (d, *J_9–10_* = 7.8 Hz, 1H, H9), 7.41 (d, *J_15–14,17–18_* = 8.4 Hz, 2H, H15, H17), 7.28–7.24 (m, 1H, H11), 7.22 (d, *J*_3–4, 7–6_ = 8.8 Hz, 2H, H3, H7), 6.89–6.85 (m, 4H, H4, H6, H14, H18), 6.53 (s, 1H, H1), 3.70 (s, 3H, H19, OMe). **^13^C NMR (100 MHz, DMSO-*d_6_*) *δ* (ppm):** 160.56 (C5), 158.7 (C8), 154.9 (C13), 149 (C12), 143 (C16), 137.2 (C10), 132.3 (C2), 128.2 (C15, C17), 126 (C3, C7), 122.7 (C9), 120.4 (C11), 115.2 (C4, C6), 113.8 (C14, C18), 80.9 (C1), 55 (C19), 33.7 (C20), 31.2 (C21). **LRMS:** (ES+, CV = 30) *m*/*z*: 349.27 [M + 2H]^+^; 348.21 [M + H]^+^; 214.12 [M − C_10_H_13_]^+^; 198.13 [M − C_10_H_13_O]^+^. **HRMS:** calcd. for C_23_H_25_NO_2_H [M + H]^+^ (348.1958); found (348.1959).

#### 2.4.5. 2-((4-chlorophenoxy)(4-methoxyphenyl)methyl)pyridine **21**

215.79 mg (1.38 mmol, 3 eq) of 4-chlorophenylboronic acid was used. After FCC, the 2-((4-chlorophenoxy)(4-methoxyphenyl)methyl)pyridine **21** was afforded as a white solid (135.2 mg) in 91% yield. **TLC:** CyHex/EtOAc: 60/40, R*_f_* = 0.7. **IR ν (cm^−1^):** 3056, 3005 (νC_sp2-H_); 2931 (ν _Csp3-H_); 2837 (ν_OMe_); 1604, 1588, 1511 and 1487 (ν_C=C_); 1229 (ν_asym C-O-C_); 1029 (ν_sym C-O-C_); 820 (*δ*_Csp2-H p-substitution_). **^1^H NMR (400 MHz, DMSO*-d_6_* ) *δ* (ppm):** 8.62 (ddd, *J_12–11_* = 4.7 Hz, *J_12–10_* = 1.9 Hz, *J_12–9_* = 0.9 Hz, 1H, H12), 7.79 (td, *J_10–9,10–11_* = 7.8 Hz, *J_10–12_* = 1.8 Hz, 1H, H10), 7.57 (d, *J_9–10_* = 7.8 Hz, 1H, H9), 7.42 (d, *J_3–4,7–6_* = 8.7 Hz 2H, H3, H7), 7.29–7.25 (m, 3H, H11, H15, H17), 7.01 (d, *J_14–15, 18–17_* = 9 Hz, 2H, H14, H18), 6.90 (d, *J_4–3, 6–7_* = 8.7 Hz, 2H, H4, H6), 6.41 (s, 1H, H1), 3.70 (s, 3H, H19, OMe). **^13^C NMR (100 MHz, DMSO-*d_6_*) *δ* (ppm):** 159.9 (C5), 158.8 (C13), 156 (C8), 149.1 (C12), 137.2 (C10), 131.7 (C2), 129.2 (C16, C18), 128.3 (C3, C7), 124.6 (C16), 122.8 (C9), 120.5 (C11), 117.6 (C14, C18), 113.8 (C4, C6), 81.23 (C1), 55.0 (C19). **LRMS:** (ES+, CV = 30) *m*/*z*: 348.10 [M + Na]^+^; 326.11 [M + H]^+^; 214.14 [M − C_6_H_4_Cl]^+^; 198.14 [M − C_6_H_4_ClO]^+^. **HRMS:** calcd. for C_20_H_16_ClNO_2_H [M + H]^+^ (326.0942); found (326.0944).

#### 2.4.6. 2-((4-bromophenoxy)(4-methoxyphenyl)methyl)pyridine **22**

277.14 mg of 4-bromophenylboronic acid (1.38 mmol, 3 eq) was added. After FCC, the 2-((4-bromophenoxy)(4-methoxyphenyl)methyl)pyridine **22** was afforded as a white solid (140 mg) in 83% yield. **TLC:** CyHex/EtOAc: 60/40, R*_f_* = 0.5. **IR ν (cm^−1^):** 3066 (ν_Csp2-H_); 2974, 2919 (ν_Csp3-H_); 2837 (ν_OMe_); 1615, 1588, 1510 and 1488 (ν_C=C_); 1228 (ν_asym C-O-C_); 1019 (ν_sym C-O-C_), 811 (*δ*_Csp2-H p-substitution_). **^1^H NMR (400 MHz, DMSO*-d_6_* ) *δ* (ppm):** 8.51 (ddd, *J_12–11_* = 4.7 Hz, *J_12–10_* = 1.9 Hz, *J_12–9_* = 0.9 Hz, 1H, H12), 7.79 (td, *J_10–9,10–11_* = 7.7 Hz, *J_10–12_* = 1.8 Hz, 1H, H10), 7.57 (d, *J_9–10_* = 7.9 Hz, 1H, H9), 7.42–7.37 (m, 4H, H3, H7, H15, H17), 7.29–7.25 (m, 1H, H11), 6.96 (d, *J_14–15, 18–17_* = 9 Hz, 2H, H14, H18), 6.89 (d, *J_4–3, 6–7_* = 8.8 Hz, 2H, H4, H6), 6.40 (s, 1H, H1), 3.70 (s, 3H, H19, OMe). **^13^C NMR (100 MHz, DMSO-*d_6_*) *δ* (ppm):** 159.9 (C13), 158.9 (C5), 156.5 (C8), 149.1 (C12), 137.2 (C10), 132.1 (C15, C17), 131.7 (C2), 128.3 (C3, C7), 122.8 (C9), 120.5 (C11), 118.2 (C14, C18), 113.9 (C4, C6), 112.4 (C16), 81.17 (C1), 55.0 (C19). **LRMS:** (ES+, CV = 30) *m*/*z*: 370.02 [M + H]^+^; 214.1 [M − C_6_H_4_Br]^+^; 198.16 [M − C_6_H_4_OBr]^+^. **HRMS:** calcd. for C_19_H_16_BrNO_2_H [M + H]^+^ (370.0437); found (370.0439).

#### 2.4.7. 2-((4-fluorophenoxy)(4-methoxyphenyl)methyl)pyridine **23**

193 mg of 4-fluorophenylboronic acid (1.38 mmol, 3 eq) was added. After FCC, the 2-((4-fluorophenoxy)(4-methoxyphenyl)methyl)pyridine **23** was afforded as a transparent oil (139 mg) in 98% yield. **TLC:** CyHex/EtOAc: 60/40, R*_f_* = 0.67. **IR ν (cm^−1^):** 3055, 3005 (ν_Csp2-H_); 2920 (ν _Csp3-H_); 2838 (ν_OMe_); 1600, 1588, 1501 and 1467 (ν_C=C_); 1242 (ν_asym C-O-C_); 1029 (ν_sym C-O-C_); 823 (*δ*_Csp2-H p-substitution_). **^1^H NMR (400 MHz, DMSO*-d_6_* ) *δ* (ppm):** 8.50 (ddd, *J_12–11_* = 4.7 Hz, *J_12–10_* = 1.9 Hz, *J_12–9_* = 0.9 Hz, 1H, H12), 7.78 (td, *J_10–9,10–11_* = 7.7 Hz, *J_10–12_* = 1.8 Hz, 1H, H10), 7.58 (d, *J_9–10_* = 7.9 Hz, 1H, H9), 7.40 (d, *J*_3–4,7–6_ = 8.7 Hz 2H, H3, H7), 7.28–7.24 (m, 1H, H11), 7.07–6.97 (m, 4H, H14, H15, H17, H18), 6.89 (d, *J_4–3, 6–7_* = 8.7 Hz, 2H, H4, H6), 6.63 (s, 1H, H1), 3.70 (s, 3H, H19, OMe). **^13^C NMR (100 MHz, DMSO-*d_6_*) *δ* (ppm):** 160.2 (C5), 158.8 (C8), 155.4 (C13), 153.5 (C16), 149.1 (C12), 137.2 (C10), 131.9 (C2), 128.3 (C3, C7), 122.8 (C9), 120.5 (C11), 117.3 (C15, C18), 115.9 (C14, C7), 113.8 (C4, C6), 81.5 (C1), 55.0 (C19). **LRMS:** (ES+, CV = 30) *m*/*z*: 311.23 [M + 2H]^+^; 310.14 [M + H]^+^; 214.12 [M − C_6_H_4_F]^+^; 198.12 [M − C_6_H_4_FO]^+^. **HRMS:** calcd. for C_19_H_16_FNO_2_H [M + H]^+^ (310.1238); found (309.2037).

#### 2.4.8. 2-((4-methoxyphenyl)(4-(trifluoromethyl)phenoxy)methyl)pyridine **24**

215.79 mg of 4-trifluorophenylboronic acid (1.38 mmol, 3 eq) was added. After FCC, the 2-((4-methoxyphenyl)(4-(trifluoromethyl)phenoxy)methyl)pyridine **24** was afforded as a yellow oil (105.9 mg) in 64% yield. **TLC:** CyHex/EtOAc: 60/40, R*_f_* = 0.6. **IR ν (cm^−1^):** 3058, 3009 (ν_Csp2-H_); 2990, 2934 (ν_Csp3-H_); 2839 (ν_OMe_); 1613, 1588, 1511 and 1468 (ν_C=C_); 1238 (ν_asym C-O-C_); 1029 (ν_sym C-O-C_); 834 (*δ*_Csp2-H p-substitution_). **^1^H NMR (400 MHz, DMSO*-d_6_* ) *δ* (ppm):** 8.53 (ddd, *J_12–11_* = 4.7 Hz, *J_12–10_* = 1.9 Hz, *J_12–9_* = 0.9 Hz, 1H, H12), 7.80 (td, *J_10–9,10–11_* = 7.7 Hz, *J_10–12_* = 1.8 Hz, 1H, H10), 7.61–7.58 (m, 3H, H9, H15, H17), 7.44 (d, *J_3–4,7–6_* = 8.7 Hz 2H, H3, H7), 7.30–7.27 (m, 1H, H11), 7.17 (d, *J_14–15, 18–17_* = 8.5 Hz, 2H, H14, H18), 6.91 (d, *J_4–3, 6–7_* = 8.8 Hz, 2H, H4, H6), 6.54 (s, 1H, H1), 3.71 (s, 3H, H19, OMe). **^13^C NMR (100 MHz, DMSO-*d_6_*) *δ* (ppm):** 160 (C13), 159.6 (C5), 158.9 (C8), 149.2 (C12), 137.3 (C10), 131.4 (C2), 128.3 (C3, C7), 126.9 (C16), 125,7 (C20), 122.9 (C9), 121.6 (C11), 120.5 (C15, C17), 116.3 (C14, C18), 113.9 (C4, C6), 81 (C1), 55.0 (C19). **LRMS:** (ES+, CV = 30) *m*/*z*: 361.19 [M + 2H]^+^; 360.12 [M + H]^+^; 214.13 [M − C_7_H_4_F_3_]^+^; 198.14 [M − C_7_H_4_F_3_O]^+^. **HRMS:** calcd. for C_20_H_16_F_3_NO_2_H [M + H]^+^ (360.1206); found (360.1207).

#### 2.4.9. 2-((4-methoxyphenyl)(4-nitrophenoxy)methyl)pyridine **25**

230 mg of 4-nitrophenylboronic acid (1.38 mmol, 3 eq) was added. After FCC, the 2-((4-methoxyphenyl)(4-nitrophenoxy)methyl)pyridine **25** was afforded as a yellow oil (151.5mg) in 98% yield. **TLC:** CyHex/EtOAc 60:40, R*_f_* = 0.5. **IR ν (cm^−1^):** 3079, 3006 (ν_Csp2-H_); 2932 (ν_Csp3-H_); 2838 (ν_OMe_); 1610, 1588, 1509 and 1467 (ν_C=C_); 1238 (ν_asym C-O-C_); 1025 (ν_sym C-O-C_); 843 (*δ*_Csp2-H p-substitution_). **^1^H NMR (400 MHz, DMSO*-d_6_* ) *δ* (ppm):** 8.54 (ddd, *J_12–11_* = 4.7 Hz, *J_12–10_* = 1.9 Hz, *J_12–9_* = 0.9 Hz, 1H, H12), 8.15 (d, *J_15–14, 17–18_* = 9.2 Hz_,_ 2H, H15, H17), 7.82 (td, *J_10–9,10–11_* = 7.8 Hz, *J_10–12_* = 1.8 Hz, 1H, H10), 7.60 (d, *J_9–10_* = 7.9 Hz, 1H, H9), 7.45 (d, *J_3–4,7–6_* = 8.7 Hz, 2H, H3, H7), 7.32–7.28 (m, 1H, H11), 7.21 (d, *J_14–15, 18–19_* = 9.2 Hz, 2H, H14, H18), 6.91 (d, *J_4–3, 6–7_* = 8.7 Hz, 2H, H4, H6), 6.64 (s, 1H, H1), 3.71 (s, 3H, H19, OMe). **^13^C NMR (100 MHz, DMSO-*d_6_*) *δ* (ppm):** 162.5 (C13), 159.1 (C5), 159 (C8), 149.3 (C12), 141.0 (C16), 137.4 (C10), 131 (C2), 128.4 (C3, C7), 125.8 (C15, C17), 123 (C9), 120.7 (C11), 116.3 (C14, C18), 114 (C4, C6), 81.4 (C1), 55.1 (C19). **LRMS:** (ES+, CV = 30) *m*/*z*: 359.09 [M + Na]^+^; 337.16 [M + H]^+^; 214.16 [M − C_6_H_4_NO2]^+^; 198.15 [M − C_6_H_4_NO_3_]^+^. **HRMS:** calcd. for C_19_H_16_N_2_O_4_H [M + H]^+^ (337.1183); found (337.1184).

#### 2.4.10. 4-((4-methoxyphenyl)(pyridin-2-yl)methoxy)-N,N dimethylaniline **26**

227.7 mg of 4-dimethylaminophenylboronic acid (1.38 mmol, 3 eq) was added. After FCC, the 4-((4-methoxyphenyl)(pyridin-2-yl)methoxy)-*N,N*-dimethylaniline **26** was afforded as a blue oil (80 mg) in 52% yield. **TLC:** CyHex/EtOAc 60:40, R*_f_* = 0.6. **IR ν (cm^−1^):** 3049 (ν_Csp2-H_); 2933, 2898 (ν_Csp3-H_); 2837 (ν_OMe_); 1610, 1589, 1509 and 1457 (ν_C=C_); 1338 (ν_C-N-C_); 1225 (ν_asym C-O-C_); 1025 (ν_sym C-O-C_); 813 (*δ*_Csp2-H p-substitution_). **^1^H NMR (400 MHz, DMSO*-d_6_* ) *δ* (ppm):** 8.49 (ddd, *J_12–11_* = 4.7 Hz, *J_12–10_* = 1.9 Hz, *J_12–9_* = 0.9 Hz, 1H, H12), 7.77 (td, *J_10–9,10–11_* = 7.6 Hz, *J_10–12_* = 1.8 Hz, 1H, H10), 7.58 (d, *J_9–10_* = 7.8 Hz, 1H, H9), 7.38 (d, *J_3–4, 7–6_* = 8.6 Hz_,_ 2H, H3, H7), 7.26–7.23 (m, 1H, H11), 6.89–6.83 (m, 4H, H6, H4, H14, H18), 6.61 (d, *J_15–14, 17–18_* = 9.1 Hz, 2H, H15, H17), 6.2 (s, 1H, H1), 3.70 (s, 3H, H19, OMe), 2.70 (s, 6H, H20). **^13^C NMR (100 MHz, DMSO-*d_6_*) *δ* (ppm):** 160.9 (C5), 158.7 (C8), 150.87 (C13), 148.9 (C12), 145.6 (C16), 137 (C10), 132.6 (C2), 128.2 (C3, C7), 122.6 (C9), 120.4 (C11), 116.8 (C14, C18), 113.9 (C15, C17), 113.7 (C4, C6), 81.7 (C1), 55 (C19), 40.9 (C20). **LRMS:** (ES+, CV = 30) *m*/*z*: 336.26 [M + 2H]^+^; 335.19 [M + H]^+^; 214.14 [M − C_8_H_10_N]^+^; 198.16 [M − C_8_H_10_NO]^+^. **HRMS:** calcd. for C_21_H_19_NO_3_H [M + H]^+^ (335.1754); found (335.1755).

#### 2.4.11. 1-(4-((4-methoxyphenyl)(pyridin-2-yl)methoxy)phenyl)ethanone **27**

189 mg of 4-acetylphenylboronic acid (1.38 mmol, 3 eq) was used. After FCC, the 1-(4-((4-methoxyphenyl)(pyridin-2-yl)methoxy)phenyl)ethanone **27** was afforded as a transparent oil (131 mg) in 78% yield. **TLC:** CyHex/EtOAc 60:40, R*_f_* = 0.3. **IR ν (cm^−1^):** 3054, 3003 (ν_Csp2-H_); 2931 (ν_Csp3-H_); 2837 (ν_OMe_); 1673 (ν_C=O_); 1604, 1580, 1507 and 1467 (ν_C=C_); 1234 (ν_asym C-O-C_); 1025 (ν_sym C-O-C_); 831 (*δ*_Csp2-H p-substitution_). **^1^H NMR (400 MHz, DMSO*-d_6_* ) *δ* (ppm):** 8.53 (ddd, *J_12–11_* = 4.7 Hz, *J_12–10_* = 1.9 Hz, *J_12–9_* = 0.9 Hz, 1H, H12), 7.86 (d, *J_15–14, 17–18_* = 8.9 Hz, 2H, H15, H17), 7.80 (td, *J_10–9,10–11_* = 7.7 Hz, *J_10–12_* = 1.8 Hz, 1H, H10), 7.58 (d, *J*_9–10_ = 7.9 Hz, 1H, H9), 7.44 (d, *J_3–4,7–6_* = 8.7 Hz 2H, H3, H7), 7.30–7.27 (m, 1H, H11), 7.09 (d, *J_14–15, 18–17_* = 8.9 Hz, 2H, H14, H18), 6.90 (d, *J_4–3, 6–7_* = 8.8 Hz, 2H, H4, H6), 6.55 (s, 1H, H1), 3.71 (s, 3H, H19, OMe), 2.46 (s, 3H, H21). **^13^C NMR (100 MHz, DMSO-*d_6_*) *δ* (ppm):** 196.2 (C20), 161 (C13), 159.7 (C5), 158.9 (C8), 149.2 (C12), 137.3 (C10), 131.5 (C2), 130.3 (C15, C17), 130.1 (C16), 128.4 (C3, C7), 122.9 (C9), 120.5 (C11), 115.6 (C14, C18), 113.9 (C4, C6), 81 (C1), 55 (C19), 26.4 (C21). **LRMS:** (ES+, CV = 30) *m*/*z*: 335.19 [M + 2H] ^+^; 214.14 [M − C_8_H_7_O]^+^; 198.14 [M − C_8_H_7_O_2_]^+^. **HRMS:** calcd. for C_21_H_19_NO_3_H [M + H]^+^ (334.1438); found (334.1439).

### 2.5. Preparation of N-oxides

#### 2.5.1. 2-(2-(4-(tert-butyl)phenyl)-1-(4-methoxyphenyl)vinyl)pyridine 1-oxide **28**

To a solution of 2-(2-(4-(tert-butyl)phenyl)-1-(4-methoxyphenyl)vinyl)pyridine **10a** (0.288 mmol, 99 mg) in anhydrous DCM was added the 3-chloroperbenzoic acid (*m-*CPBA) (0.288 mmol, 50 mg, 1 eq) under argon. The reaction mixture was stirred at room temperature during 3 h. After reaction completion, a solution of KOH (40%) was added to the mixture until pH 8–9. The mixture was washed then with water and extracted with DCM. The organic layer was dried under anhydrous MgSO_4_, filtered and concentrated. The crude residue was purified by FCC on silica gel (DCM/MeOH.NH_3_ (7N): 98/2) to afford the 2-(2-(4-(tert-butyl)phenyl)-1-(4-methoxyphenyl)vinyl)pyridine 1-oxide **28** as a white solid in 53% yield. **TLC:** DCM/MeOH: 95/5, R*_f_* = 0.25. **IR ν (cm^−1^):** 3676 (ν_OH_); (ν_Csp2-H_); 2986 (ν_Csp3-H)_; 2983 (ν_O-Me_); 1405 (ν_C=C_); 1254 (ν_C-O_); 894 (*δ*_Csp2-H p-substitution_). **^1^H NMR (400 MHz, CDCl_3_) *δ* (ppm):** 8.39 (d, J= 6.5 Hz, 1H, H12 ), 7.31–7.27 (m, 1H, H9), 7.7.23 (s, 1H, H, H13), 7.20–7.16 (m, 6H, H10, H11, H3, H7, H15, H19), 6.98 (d, *J_15–16_ = J _18–19_* = 8.36 Hz, 2H, H16, H18), 6.86 (d, *J_4–3_ = J_6–7_* = 8.8 Hz, 2H, H4, H6), 3.7 (s, 3H, H20, OMe), 1.24 (s, 9H, H22). **^13^C NMR (100 MHz, DMSO*-d_6_*) δ (ppm):** 159.64 (C5), 150.88 (C8), 140.47 (C12), 133.35 (C1), 133.27 (C10), 131.53 (C17), 130.38 (C14), 129.00 (C2), 128.50 (C4, C6), 127.20 (C3, C7), 125.75 (C15, C19), 125.42 (C13), 125.29 (C9), 114.31 (C16, C18), 55.47 (C20), 34.69 C21), 31.32 (C22).

#### 2.5.2. 2-(1-(4-methoxyphenyl)-2-(4-(trifluoromethyl)phenyl)vinyl)pyridine 1-oxide **29**

To a solution of 2-(2-(4-(tert-butyl)phenyl)-1-(4-methoxyphenyl)vinyl)pyridine **10a** (0.288 mmol, 99 mg) in anhydrous DCM was added the *m-*CPBA (0.288 mmol, 50 mg, 1 eq) under argon. The reaction mixture was stirred at room temperature during 3 h. After reaction completion, a solution of KOH (40%) was added to the mixture until pH 8–9. The mixture was washed then with water and extracted with DCM. The organic layer was dried under anhydrous MgSO_4_, filtered and concentrated. The crude residue was purified by FCC on silica gel (DCM/MeOH.NH_3_ (7N): 98/2) to afford the 2-(2-(4-(*tert-*butyl)phenyl)-1-(4-methoxyphenyl)vinyl)pyridine 1-oxide **28** as a white solid in 53% yield. **TLC:** DCM/MeOH: 95/5, R*_f_* = 0.25. **IR ν (cm^−1^):** 3676 (ν_OH_); (ν_Csp2-H_); 2986 (ν_Csp3-H)_; 2983 (ν_O-Me_); 1405 (ν_C=C_); 1254 (ν_C-O_); 894 (*δ*_Csp2-H p-substitution_). **^1^H NMR (400 MHz, CDCl_3_) *δ* (ppm):** 8.39 (d, *J* = 6.5 Hz, 1H, H12), 7.31–7.27 (m, 1H, H9), 7.7.23 (s, 1H, H, H13), 7.20–7.16 (m, 6H, H10, H11, H3, H7, H15, H19), 6.98 (d, *J_15–16_ = J _18–19_* = 8.36 Hz, 2H, H16, H18), 6.86 (d, *J_4–3_ = J_6–7_* = 8.8 Hz, 2H, H4, H6), 3.7 (s, 3H, H20, OMe), 1.24 (s, 9H, H22). **^13^C NMR (100 MHz, DMSO*-d_6_*) δ (ppm):** 159.64 (C5), 150.88 (C8), 140.47 (C12), 133.35 (C1), 133.27 (C10), 131.53 (C17), 130.38 (C14), 129.00 (C2), 128.50 (C4, C6), 127.20 (C3, C7), 125.75 (C15, C19), 125.42 (C13), 125.29 (C9), 114.31 (C16, C18), 55.47 (C20), 34.69 C21), 31.32 (C22).

#### 2.5.3. 2-((4-methoxyphenyl)(phenoxy)methyl)pyridine 1-oxide **30**

To a solution of 2-((4-methoxyphenyl)(phenoxy)methyl)pyridine **17** (0.288 mmol, 84 mg) in anhydrous DCM was added the *m-*CPBA (1.55 mmol, 348 mg, 5.3 eq) under argon. The reaction mixture was stirred at room temperature during 3 h. After reaction completion, a solution of KOH (40%) was added to the mixture until pH 8–9. The mixture was washed then with water and extracted with DCM. The organic layer was dried under anhydrous MgSO_4_, filtered and concentrated. The crude residue was purified by FCC on silica gel (DCM/MeOH.NH_3_ (7N): 98/2) to afford the 2-((4-methoxyphenyl)(phenoxy)methyl)pyridine 1-oxide **81c** as a white oil in 99% yield (87.7 mg). **TLC:** DCM/MeOH: 95/5, R*_f_* = 0.2. **IR ν (cm^−1^):** 3062 (ν_Csp2-H_); 2935 (ν_Csp3-H_); 2836 (ν_OMe_); 1587, 1511 and 1488 (ν_C=C_); 1286 (ν_N-O_), 1221 (ν_C-O_); 751 (*δ*_Csp2-H p-substitution_). **^1^H NMR (400 MHz, DMSO-*d_6_*) *δ* (ppm):** 8.28 (ddd, *J_12–11_* = 4.7 Hz, *J_12–10_* = 1.9 Hz, *J_12–9_* = 0.9 Hz, 1H, H12), 7.67 (m, 1H, H9), 7.46 (d, *J_3–4, 7–6_* = 8.7 Hz, 2H, H3, H7), 7.37 (m, 2H, H10, H11), 7.29–7.23 (m, 2H, H15, H17), 6.96–6.90 (m, 5H, H4, H6, H14, H16, H18), 6.85 (s, 1H, H1), 3.73 (s, 3H, H19, OMe). **^13^C NMR (100 MHz, DMSO-*d_6_*) *δ* (ppm):** 159.68 (C13), 157.17 (C5), 150.40 (C8), 139.78 (C12), 130.16 (C15, C17), 129.89 (C2), 129.33 (C3, C7), 126.13 (C10), 125.89 (C11), 124.17 (C16), 121.89 (C9), 115.89 (C14, C18), 114.28 (C4, C6), 73.52 (C1), 55.59 (C19). **LRMS:** (ES+, CV = 30) *m*/*z*: 637.31 [2M + Na]^+^; 330.17 [M + Na]^+^; 308.24 [M + H]^+^; 214.16 [M − C_6_H_4_O]^+^. **HRMS:** calcd. for C_19_H_17_NO_3_Na [M + Na]^+^ (330.1101); found (330.1101).

#### 2.5.4. 2-((4-bromophenoxy)(4-methoxyphenyl)methyl)pyridine 1-oxide **31**

To a solution of 2-((4-bromophenoxy)(4-methoxyphenyl)methyl)pyridine **22** (0.389 mmol, 144 mg) in anhydrous DCM was added the *m-*CPBA (2 mmol, 469 mg, 5 eq) under argon. The reaction mixture was stirred at room temperature during 3 h. After reaction completion, a solution of KOH (40%) was added to the mixture until pH 8–9. The mixture was washed with water and extracted with DCM. The organic layer was dried under anhydrous MgSO_4_, filtered and concentrated. The crude residue was purified by FCC on silica gel (DCM/MeOH NH_3_, 7N, 2 to 10% of MeOH.NH_3_, (7N)) to afford the 2-((4-bromophenoxy)(4-methoxyphenyl) methyl)pyridine 1-oxide **31** as a brown solid in 88% yield (132 mg). **TLC:** DCM/MeOH: 95/5, R*_f_* = 0.2. **IR ν (cm^−1^):** 3065, 3019 (ν_Csp2-H_); 2931 (ν_Csp3-H_); 2837 (ν_OMe_); 1610, 1582 and 1511 (ν_C=C_); 1278 (ν_N-O_), 1225 (ν_C-O_); 764 (*δ*_Csp2-H p-substitution_). **^1^H NMR (400 MHz, DMSO*-d_6_*) *δ* (ppm):** 8.29 (ddd, *J_12–11_* = 4.7 Hz, *J_12–10_* = 1.9 Hz, *J_12–9_* = 0.9 Hz, 1H, H12), 7.68–7.64 (m, 1H, H9), 7.48–7.36 (m, 6H, H3, H7, H15, H17, H10, H11), 6.93–6.90 (m, 4H, H6, H4, H14, H18), 6.84 (s, 1H, H1), 3.73 (s, 3H, H19, OMe). **^13^C NMR (100 MHz, DMSO*-d_6_*) *δ* (ppm):** 159.30 (C13), 156.00 (C5), 149.53 (C8), 139.36 (C12), 132.36 (C15, C17), 132.03 (C2), 128.95 (C3, C7), 125.70 (C10), 125.55 (C11) 123.74 (C9), 117.87 (C14, C18), 112.9 (C16), 113.86 (C4, C6), 73.49 (C1), 55.14 (C19). **LRMS:** (ES+, CV = 30) *m*/*z*: 795.02 [2M + Na]^+^; 386.10 [M]^+^; 370.24 [M − O]^+^; 304.93 [M − Br]^+^ 230.24 [M − C_6_H_4_Br]^+^.

### 2.6. Bioassays

#### 2.6.1. Materials

Dulbecco’s modified eagle medium (DMEM, Roswell Park Memorial Institute medium (RPMI) 1640 medium, fetal bovine serum (FBS), L-glutamine and penicillin-streptomycin were purchased from Gibco BRL—Fisher Scientific (Cergy-Pontoise, France). Trypsin was purchased from Pan-Biotech (Dutscher, Bernolsheim, France). 3-(4,5-dimethylthiazol-2-yl)-2,5-diphenyltetrazolium bromide (MTT), DMSO and β-actin antibody were obtained from Sigma-Aldrich—Merck (Saint-Quentin-Fallavier, France). Caspase-3, cleaved caspase-3, poly-ADP-ribose polymerase (PARP) as well as P-Akt, Akt, P-ERK, ERK, P-p38, p-38 antibodies and goat anti-rabbit IgG secondary antibody conjugated to horseradish peroxidase (HRP) were acquired from Cell Signaling Technology—Ozyme (Saint-Quentin-en-Yvelines, France). Rabbit anti-mouse IgG, IgM (H+L) secondary antibody conjugated to HRP was obtained from Invitrogen—Fisher Scientific and PVDF membranes from GE Healthcare Life Science—Fisher Scientific. Immobilon Western Chemiluminescent HRP Substrate Cell death was acquired from Millipore—Fisher Scientific. Protease inhibitors (Complete™ Mini) and detection enzyme-linked immunosorbent assay^PLUS^ (ELISA) were purchased from Roche Diagnostics—Merck (Lyon, France).

#### 2.6.2. Cells Lines, Cell Culture and Treatment

Human CRC HCT116 and HT-29 adherent cell lines were purchased from the American Type Culture Collection (ATCC—LGC Standards, Molsheim, France). We chose these human CRC cell lines because they are of different stages in order to evaluate possible resistance to our treatments: the HCT116 CRC line was isolated from a stage I colorectal carcinoma of an adult male. The HT-29 CRC line was derived from a stage II colorectal adenocarcinoma from a 44-year-old woman.

Cells were grown in DMEM medium for HT-29 cells and RPMI 1640 medium for HCT116 cells, supplemented with 10% FBS, 1% L-glutamine and 100 U/mL penicillin and 100 μg/mL streptomycin. Cultures were maintained in a humidified atmosphere containing 5% CO_2_ at 37 °C. Stock solutions of each compound were used at 10^−2^ M in DMSO and then diluted in culture medium to obtain the appropriate final concentrations. The same amount of vehicle (percentage of DMSO did not exceed 0.5%) was added to control cells. L929 cell line is a non-cancer cell line derived from L-strain (L cells) and has been grown in the Peptinov laboratory for years. L929 cells are murine adherent fibroblasts from subcutaneous connective tissue (areolar and adipose tissues). Cells were grown in DMEM medium supplemented with 10% FBS, 1% L-glutamine and 100 U/mL penicillin and 100 µg/mL streptomycin and maintained in a humidified incubator at 37 °C, 5% CO_2_. When confluence is at 80%, cells are trypsinyzed for 3 min and diluted in fresh medium. Each batch of cells is kept for 15 passages before being discarded and a new batch thawed.

#### 2.6.3. Cell Metabolic Activity

All compounds were tested on the metabolic activity of the cells using the MTT colorimetric assay [29]. Briefly, cells were seeded in 96-well microplates at 8 × 10^3^ cells/well for human CRC HT-29 cells and 5 × 10^3^ cells/well for human CRC HCT116 cells and grown for 24 h in appropriate culture medium prior to exposure or not to compounds (**6–27**) with concentration ranges from 1 to 50 µM. After 48 h of treatment, MTT (5 g/L in Phosphate-buffered saline (PBS)) was added and incubated for another 3 h. The MTT was then removed from the wells and 100 µL/well of DMSO were added to dissolve formazan. The optical density was detected with a microplate reader (Thermoscientific, Multiskan FC) at 550 nm and cell viability was expressed as a percentage of each treatment condition compared to control cells. IC_50_ values were calculated for all compounds from the dose–response curve.

Cell viability of L929 cell line was evaluated in presence of synthesized compounds on. Briefly, cells were trypsinyzed and seeded in 96-well microplates at 4 × 10^4^ cells/well for 24 h in a humidified incubator at 37 °C, 5% CO_2_. The following day, 100 µL of a mix containing serial diluted compounds, in constant 0.5% DMSO, were added to the cells and plates were incubated for 24 h in a humidified incubator at 37 °C, 5% CO_2_. 100 µL of a mix containing 0.5% of DMSO was added to untreated cells (control cells). After removing the mix, 100 µL of 0.5 mg/mL of MTT were added to the cells and plates were incubated for 2 h in an humidified incubator at 37 °C, 5% CO_2_. MTT was then removed and 200 µL of DMSO were added to the wells to dissolve formazan crystals. Optical density was then read with a spectrophotometer (Multiskan, Fisher Scientific, Illkirch, France) at 560 nm. Cell viability was expressed as percentages compared to untreated cells.

#### 2.6.4. Protein Extraction and Western Blot Analysis

Human CRC HT-29 and HCT116 cells were treated or not with the determined IC_50_ values of compounds (**12a, 10a, 10b**) for indicated times (6, 12, 24 and 48h) and then harvested with trypsin. For total protein extraction, collected samples of each condition were washed with PBS. Then, the total cell pool was centrifuged at 200× *g* for 5 min at 4 °C and homogenized in RIPA lysis buffer (50 mM HEPES, pH 7.5, 150 mM NaCl, 1% sodium deoxycholate, 1% NP-40, 0.1% Sodium Dodecyl Sulfate (SDS), 20 mg/mL of aprotinin) containing protease inhibitors according to the manufacturer’s instructions as previously described [30]. Proteins (60 µg) were separated on 12.5% SDS-PAGE gels and transferred to polyvinylidene difluoride (PVDF) membranes. Membranes were probed with respective human antibodies against caspase-3, cleaved caspase-3, PARP and Akt, ERK, p38 MAP Kinases and its phosphorylated forms according to the manufacturer’s instructions. After incubation with appropriate secondary antibodies, blots were developed using the «Immobilon Western » substrate following the manufacturer’s protocol and G:BOX system (Syngene, Ozyme). Membranes were then reblotted with human anti-β-actin used as a loading control.

#### 2.6.5. Apoptosis Quantification by DNA Fragmentation Analysis

Human CRC HT-29 and HCT116 cells were treated or not with the determined IC_50_ values of compounds (**12a, 10a, 10b**) for 24 and 48 h and then harvested with trypsin. Histone release from the nucleus during apoptosis was analyzed using the Cell Death Detection ELISA^PLUS^ as previously described [6]. Cytosol extracts from 10^5^ cells of each condition were obtained and DNA fragmentation was measured according to the manufacturer’s protocol. Results were reported as n-fold compared to control cells.

#### 2.6.6. Statistical Analysis

Data are expressed as the arithmetic means ± standard error of the mean (SEM) of at least three separate experiments. Statistical significance was evaluated by the two-tailed unpaired Student’s t-test and expressed as: * *p* < 0.05; ** *p* < 0.01 and *** *p* < 0.001.

### 2.7. Molecular Modeling

#### 2.7.1. Protein and Compounds Preparation

The protein structures of the five potential targets were extracted from the Protein Data Bank (PDB): [31] AKT (PDB ID: 6S9W), ERK-1 (PDB ID: 4QTB), ERK-2 (PDB ID: 6SLG), PARP (PDB ID: 4ZZZ) and caspase-3 (PDB ID: 6CKZ). The structure of ERK-2 was superimposed on the structure of ERK-1 to allow comparison of the binding modes obtained on these two isoforms. All the structures were prepared using the DockPrep tool from UCSF Chimera [32] and MGL tools [33].

Three dimensional structures of compounds **12a** and **10a** were generated using iCon [34], the LigandScout v.4.3. conformer generator (defaults settings of the BEST option were used, except for the maximum number of conformations that was setted to 2000). Compounds were protonated at pH 7.4 using the cxcalc plugin of the Chemaxon suite [35] and converted in .pdbqt format.

#### 2.7.2. Docking Study and Protein-Ligand Interactions Analysis

For all the five target, the docking study was conducted using smina and the vinardo scoring function [36]. For each target, a search space was defined with a size of 20 Å × 20 Å × 20 Å and the following x, y, z grid center coordinates: −12.729, −15.248, 13.193 for AKT, 36.683, −54.826, 49.926 for ERK-1 and ERK-2, 63.412, 6.484, 9.593 for PARP and 39.023, 11.547, 71.322 for caspase-3.

The predicted binding mode of each ligand in each binding site was analyzed using the Protein-Ligand Interaction (PLIP) webserver.

#### 2.7.3. ADME Profile and Drug-Likeness

Predictions of ADME properties and drug-likeness for the most promising compounds were conducted in SCHRÖDINGER Maestro v11.9 software, using the Molecular Properties Panel from QikProp v5.9 platform [37]. Structure Minimization was performed using Force field OPLS3e, [38] the Polak-Ribier method of Conjugate Gradient (PRCG), with a convergence threshold of 0.05 and a maximum of 2500 iterations.

## 3. Results and Discussion

### 3.1. Chemistry

The general synthetic pathway followed to synthesize two series of functionalized DAM: olefinic and aryloxy is shown in Figure 1. Details about the synthetic protocol and chemical characterization of all intermediates are given in the Appendix A.

First, the synthesis of carbinol **3** was performed from bromopyridine **1** and anisaldehyde **2** by a bromine-magnesium exchange using isopropylmagnesium chloride in tetrahydrofuran at room temperature [39]. The reaction was also performed by a bromine-lithium exchange following the procedure of Seto et al. [40]. Nevertheless, this latter protocol led to only 22% of the corresponding carbinol and to a by-product **4** not previously described in the literature. The corresponding arylketone **5** was synthesized from carbinol **3** in excellent yield (98%) via a base-promoted aerobic oxidation using air as a free and clean oxidant [41].

The key step to obtain the desired olefinic diarylmethanes involved a McMurry cross-coupling reaction between the aryllketone **5** and the corresponding *para-*substituted aromatic aldehyde in presence of TiCl_4/_Zn in THF. After the *in situ* formation of the catalytic entity (TiCl_2_) by the reduction of TiCl_4_ using Zn in THF at reflux for 2 h, an equimolar mixture of the arylketone **5** and the benzaldehyde **2** in THF was added dropwise to the reaction medium. Depending on the aromatic aldehyde used, containing electron-withdrawing, electron-donating and halogen groups, the reaction time varies between 10 and 30 min after the addition of reagents. The expected DAM were obtained in two separable *E* and *Z* isomers except for compound **11** (Figure 1). No selectivity was observed in the formation of major isomers. The *Z* and *E* isomers were easily distinguished by NMR analysis. Thus, it is noted that the chemical shift of the olefinic proton in NMR for *E* isomer is higher than for *Z* isomer (7.7 ppm for *E* isomer versus 7.0 ppm for *Z* isomer).

As a continuation of the medicinal chemistry program to obtain new DAM, we were also interested in the synthesis of the bioisosteric analogues (**13–17**). To achieve this purpose and after optimization, a set of aryloxyDAM were synthesized from carbinol **3** and the corresponding *para* functionalized phenylboronic acids (**25**). The reaction was performed in presence of pyridine and copper (II) diacetate in DCM at room temperature following the procedure of Sui et al. for the *O-*arylation of lactose [42]. The desired aryloxyDAM (**17–27**) were obtained in goods yields ranging between 64–98%.

The introduction of the *N*-oxide moiety has been highly considered in medicinal chemistry programs. Indeed, *N*-oxides provide interesting physicochemical properties such as improved solubility and the capability to increase affinity with receptor sites [28]. Considering these arguments and based on biological results (Section 2.2), the two *Z* isomers of the olefinic DAM **10a** and **12a** as well as the two aryloxyDAM **17** and **22**, were *N-*oxidized using *m-*chloroperbenzoic acid (*m-*CPBA) in dichloromethane at room temperature.

For the olefinic compounds **10a** and **12a**, the oxidation was performed using only 1 equivalent of *m-*CPBA in order to avoid undesirable oxidation of the double bond. However, for the aryloxyDAM **17** and **22**, as these derivatives do not present any other oxidizable site, 5 equivalents of *m-*CPBA were used. The olefinic oxidized compounds were obtained in yields of 45% and 50%, respectively. Furthermore, the corresponding pyridine *N-*oxides of aryloxyDAM are obtained in 99% and 88% yields (Figure 2).

### 3.2. Biological Evaluation

#### 3.2.1. Cell Viability, Cell Proliferation Inhibition, and IC_50_ Determination

First, cell viability was evaluated using MTT assay. Eighteen olefinic DAM and eleven aryloxyDAM previously synthesized) were evaluated on human CRC cell lines HT-29 and HCT116 at 50 µM for 48 h. In this screening, olefinic DAM series displayed a higher cell viability proliferation inhibition than their aryloxyDAM analogues. In addition, for two series HT-29 cells seems to be more sensitive to the compounds than HCT116 cells. This observation is more pronounced for olefinic DAM (Figure 3 and Figure 4).

Eleven olefinic DAM (**6a, 7a-b, 8a, 9b, 10a-b, 12a, 13a, 14a, 15a**) displayed a cell viability lower of 50% for HT-29 and while only six compounds of this series (**6a, 10a, 12a, 13a, 13b, 14a**) displayed cell viability lower of 50% for HCT116 (Figure 3). For the aryloxy DAM series, only compound **22** for HT-29 and HCT116 cells displayed a cell viability lower of 50% (Figure 4).

The determination of a median inhibitory concentration (IC_50_) at 1 to 50 µM (1, 10, 20, 30, 40 and 50 µM) was performed at 48 h on the two human CRC cell lines used for compounds with cell viability <50% (Table 1).

The obtained results highlight that only five compounds displayed an IC_50_ lower than 30 µM (**10a-b, 12a, 13a, 14a**) for HT-29 and three compounds (**10a, 12a, 13b**) displayed an IC_50_ below 35 µM for HCT116. These results allowed us to establish the first elements of the structure-activity relationships (see Section 3.2.2).

#### 3.2.2. Structure Activity Relationship Considerations

The cell proliferation inhibition induced by the DAM allowed a comprehensive structure-activity relationship (SAR) analysis. We considered the spatial configuration of either *E* or Z, the impact of olefinic carbon or the oxygen atom, the nature of the functionalization and the influence of *N*-oxide moiety.

In the olefinic DAM series, compounds with *Z* configuration displayed better antiproliferative activity compared to their corresponding *E* analogues except for compounds functionalized with bulky alkyl groups (isopropyl **9b vs. 9a** and *tert-*butyl **10b vs. 10a)** on human CRC HT-29 cells. The *Z* isomers containing Cl, Br, and CF_3_ groups (**14a, 13a and 12a**) as well as the two *Z* and *E* isomers bearing the *tert-*butyl group (**10a-b**) showed a better antiproliferative activity on the human CRC HT-29 cell line with IC_50_ values of 28.48, 24.03, 23.02, 25.70 and 25.15 µM, respectively. The brominated *E* isomer (**13b**), the *Z* isomer bearing a CF_3_ moiety (**12a**) as well as a *Z* isomer bearing a *tert-*butyl group (**10a**) displayed the best antiproliferative activity on the human CRC HCT116 cell line with IC_50_ of 26.13, 31.44 and 33.61 µM, respectively (Table 1).

These results suggested that in a general, *Z* isomers have a more interesting antiproliferative activity than the *E* isomers. These results are in agreement with those described in the literature for ferrocenyl derivatives of tamoxifen with IC_50_ = 11 µM for the *Z* isomer and IC_50_ = 60 µM for the *E* isomer [43].

Concerning the aryloxy DAM series, most of the compounds were found to be inactive towards the cell proliferation inhibition at concentrations lower than or equal to 50 µM. Only the brominated derivative **22** showed a potential antiproliferative activity on the two human CRC cell lines evaluated. These results suggested that the introduction of an oxygen atom decreases activity. Thus, this structural modification did not appear to be crucial for the antiproliferative activity on the human CRC cell lines evaluated (Table 1).

To study the influence of the introduction of an *N*-oxide pyridine motif, compounds **10a** and **12a** from the olefinic series and compounds **17** and **22** from the aryloxyDAM series were *N*-oxidized. The molecules were selected based on their antiproliferative activity and an inactive compound **17** was also selected in order to compare. The synthesized *N-*oxides were also tested at a concentration between 1 to 50 µM for 48 h.

For two series, the results displayed that pyridine *N*-oxides induce a loss of antiproliferative activity on both human CRC cell lines at a concentration of up to 50 µM. These results could suggest that non-substituted nitrogen atom in the pyridine ring is required for the biological activity.

#### 3.2.3. Mechanism of Action Investigation

The compounds **10a, 10b** and **12a,** that showed the best biological activity were selected for further investigation of the mechanism of action. To elucidate the potential target of the antiproliferative activity on the human CRC cell lines, the study of some anti-apoptotic cell survival signaling pathways (phospho-ERK, phospho-Akt) and apoptotic signaling pathways (phospho-p38) were performed. In addition, the evaluation of pro-apoptotic markers (caspase-3 and PARP cleavage, DNA fragmentation) was also carried out.

*Evaluation of pro-apoptotic markers of the survival and apoptosis signaling pathways* was performed to complete the study of the mechanism of action.

P-Akt, Akt, P-ERK, ERK, P-p38, p-38 sourced from human CRC HCT116 cells line were investigated. These cells were treated or not with IC_50_ values of compounds **12a** and **10a** for 6 and 12 h. Total lysates were collected and expression of Akt, ERK and p38 MAPK proteins and their phosphorylated forms were determined by Western blot analysis.

Protein kinase B (Akt), is a protein involved in the cell death and survival process, playing a pivotal role in several interconnected cell signaling mechanisms ultimately engaged in cell metabolism, growth and division, apoptosis suppression and angiogenesis. Once phosphorylated, this protein generates consequently the P-Akt (phosphorylated Akt) which ultimately participates in the process of oxidative stress and plays a prognostic role in cancer. The inhibition of the Akt as well as the signaling pathway for its phosphorylation prevents cell regeneration and thereby causes cell death. Analyzing the effects of compounds **12a** and **10a** on this Akt and the P-Akt, we observed that P-Akt is downregulated after 6 h of treatment with **12a,** and this inhibition effect is enhanced at 12 h. This inhibition is also observed with **10a** but this effect is not time-dependent as the level of P-Akt expression remains the same between 6 and 12 h (Figure 5).

Extracellular signal-regulated kinases (ERKs), are member of the mitogen-activated protein kinase (MAPK) involved in a series of physiological processes, such as the regulation of cell survival and proliferation as well as cell differentiation [44,45,46]. Colon tumor epithelial cells are dependent on mitogen activated protein kinase (MAPK) p38 for proliferation and survival [47,48]. We investigated the role of ERK and p38 in the observed antiproliferative effect. The results showed that ERK appeared unaltered compared to the control. On the other hand, compounds **12a** and **10a** drastically decreased ERK phosphorylation as early as 6 h of treatment, with the latter remaining at the same level of expression after 12 h of treatment with compound **10a**. In addition, no p38 activation was observed, suggesting that compounds **12a** and **10a** do not influence this pro-apoptotic signaling pathway (Figure 5).


*Evaluation of pro-apoptotic markers.*


Once the DNA in cancer cells is fragmented, enzymes such as poly(ADP-ribose) polymerase (PARP), involved in DNA reparation and consequently regeneration of the cancer cells, are mutated or inactivated by a pro-apoptotic process. Likewise, caspase-3 is an apoptosis related protein that is involved and activated during apoptosis [49]. In order to determine whether the inhibition of human CRC HCT116 and HT-29 cells line indeed affected the DNA reparation, the potential of the compounds **10a**, **10b** and **12a** to inhibit these apoptosis related proteins PARP and caspase-3 were analyzed [50,51].

Cells were treated or not for 24 and 48 h, with IC_50_ values of compounds **10a**, **10b** and **12a**, for human CRC HT-29 cells and compounds **10a** and **12a** for human CRC HCT116 cells. Total lysates were collected, and expression of apoptosis-related proteins was determined by Western blot analysis.

Only compound **12a** showed inhibition of both pro-apoptotic proteins. Treatment of human CRC HT-29 cells with the compound **12a** showed a high expression of the cleaved form of caspase-3 within 48 h of treatment (Figure 6A). Similarly, the treatment of human CRC HCT116 cells, provides the cleavage of caspase-3 over 24 h of treatment and its expression is enhanced after 48 h (Figure 6B).

In human HT-29 cells, the cleaved PARP expression is slightly enhanced for the compound **12a** within 24 h and turns overexpressed at 48 h. However, for compounds **10a** and **10b** native PARP remains constant and cleaved PARP has never been expressed either at 24 h or 48 h (Figure 6A). In human CRC HCT116 cells, compound **12a** induces a significant increase of cleaved PARP at 24 and 48 h (Figure 6B).

*DNA fragmentation* capacity was studied on human CRC HT-29 and HCT116 cell lines in two time points at 24 and 48 h. The compounds **10a**, **10b** and **12a** were evaluated at 25.70, 25.15 and 23.02, µM, respectively (IC_50_ values) on human CRC HT-29 cells. For human CRC HCT116 cells, only **12a** and **10a** were evaluated. DNA fragmentation was quantified by ELISA. The results expressed as n-fold compared to control and are shown in Figure 7.

On human CRC HT-29 cells, compounds **10a** and **10b** showed a slight DNA fragmentation compared to the control. However, significant fragmentation was observed with compound **12a** at 24 and 48 h, 2.90- and 3.25-fold, respectively compared to the control (Figure 7A). On human CRC HCT116 cells, no significant DNA fragmentation was observed after 24 h showing only 1.46-fold with compound **12a** compared to the control. However, very high fragmentation of the DNA was observed at 48 h showing 6.97-fold compared to the control as illustrated in Figure 7B. It can be noticed that compound **12a** has an immediate effect on human CRC HT-29 cells while on HCT116 cells, the effect is more delayed and is observed only at 48 h.

#### 3.2.4. Normal Cell Line Viability

Cell viability of L929 cell line (murine fibroblasts) was evaluated in presence of synthesized compounds using the MTT colorimetric assay. At the concentrations evaluated (between 0.78 and 100 µM), all compounds have no effect on the cell viability of these normal cells. The figures and graphs concerning these results are reported in the Appendix A.

### 3.3. In Silico Studies

#### 3.3.1. Molecular Modeling

In order to study the potential binding mode of compounds **10a** and **12a** in the biological targets previously identified, a docking approach was used.

For AKT (PDB ID: 6S9W), the predicted binding modes obtained for compounds **10a** and **12a** were quite identical and also very similar to a part of the co-crystallized ligand (Figure 8A). Indeed, our compounds were predicted to share similar hydrophobic interactions with the residues of the AKT binding site (with W80, L264, V270 and Y272) and π- stacking (W80), than the co-crystallized ligand (Figure 8B). Additionally, to these shared interactions, our compounds **10a** and **12a** are predicted to establish a hydrogen bond (HB) with K268.

We also evaluated the potential binding mode of our compounds **10a** and **12a** in the two isoforms of the ERK proteins, namely ERK-1 and ERK-2. The binding site of these two proteins are highly similar, however the predicted binding mode in these proteins were dissimilar (Figure 9).

In the ERK-1 binding site (PDB ID: 4QTB), our compounds are predicted to present different binding modes (Figure 10A) but with some similarities. Indeed, compounds **10a** and **12a** are both predicted to be able to establish hydrophobic interactions with ERK-1 residues, among which Y53, K71 and L173, are also involved in hydrophobic interactions with the 4QTB co-crystallized ligand. Additionally, compound **12a** predicted binding mode includes a halogen bond with the ERK-1 binding site residue E88, which is engaged in a salt bridge with the 4QTB co-crystallized ligand (Figure 10B). For compound **10a**, additional interactions are π-stacking with Y53 and π-cation interaction with K71 (Figure 10C).

In the ERK-2 binding site (PDB ID: 6SLG), our compounds are predicted to interact in a similar way, but differently from both the co-crystallized ligand (Figure 11A) and as previously mentioned, the predicted ERK-1 binding modes. Our compounds are predicted to interact with the ERK-2 binding sites (Figure 11B) through numerous hydrophobic interactions (with residues A18, Y19, V22, K37, I39 and D150) and hydrogen bonds (with residues Y19, K37 and R50).

The predicted binding modes of compounds **12a** and **10a** are very similar in the PARP binding site (4DZZ) but they are not superimposed with the co-crystallized ligand NMS-P118 due to a lack of structural similarity (Figure 12A). However, our compounds shared hydrophobic interactions (with residues Y896 and Y907) and π-stacking (with Y907) with the reference NMS-P118/PARP complex (Figure 12B). Moreover, our compounds are predicted to establish additional hydrophobic interactions (with Y889), π-stacking (with 896Y and 889Y) and HB (with M890 vs. E863 in the reference NMS-P118/PARP complex).

The caspase-3 binding site is formed by different sub-pockets. Our compounds **12a** and **10a** are less flexible than the co-crystallized ligands of caspase-3 and are thus predicted to establish interactions with different subpockets than other co-crystallized ligands. Nevertheless, the pyridine moiety of our compounds is predicted to be located in a similar hydrophobic pocket than part of a few co-crystallized ligands of caspase-3 (Figure 13A). Compounds **10a** and **12a** are predicted to establish both π-stacking and hydrophilic interactions with residues Y204, W206 and F256 of the caspase-3 binding site (Figure 13B).

The predicted binding modes of our compounds in the AKT, ERK-1 and ERK-2, PARP and caspase-3 binding sites complement and support the biological results obtained for these targets. According to the analysis of these predicted binding modes (and also the docking scores), the most promising biological target for compounds **10a** and **12a** seems to be the AKT protein.

#### 3.3.2. Prediction of ADME Properties and Druglikeness

The prediction of several properties and molecular descriptors allowed us to suggest the ADME profile and drug-likeness of the three promising drug candidates **10a**, **10b**, and **12a** (Figure 14).

Most of the physicochemical properties of the three selected candidates were within the recommended ranges. However, low aqueous solubility (PlogS) and octanol/water partition coefficient (PlogPo/w) values outside the recommended limit for the compounds were predicted. The ADME properties that refer to the capacity to cross the blood–brain barrier (PlogBB and PPMDCK), the intestinal barrier (PPCaco), or to bind to human serum albumin PlogKhsa), as well as the predicted human oral absorption (HOA%) were all predicted as good for the three compounds. Only the skin permeability (PlogKp) was not predicted within the values considered recommended. Finally, several criteria (i.e., stars, RuleOf3, and RuleOf5) demonstrated the drug-likeness compliance of the studied molecules.

## 4. Conclusions

In summary, two original series of DAM compounds, olefinic DAM and aryloxyDAM, were straightforwardly synthetized using efficient synthetic strategies, characterized, and biologically evaluated. The effects of the novel 33 DAM derivatives (18 olefinic DAM and 11 aryloxyDAM and 4 *N*-oxides derivatives) were evaluated in vitro on human CRC cell lines HT-29 and HT116. The cell-based bioassays revealed that compounds **10a**, **10b** and **12a** of the olefin series decreased the viability of cancer cells. The *Z* isomers seem to be more active than their *E* analogues and the pyridine cycle is important for the activity. The mechanism of action of compounds **10a**, **10b**, and **12a** was investigated. We can conclude that compound **12a** has a very interesting anti-cancer potential. Thus, the effects of **12a** in inducing caspase-3 cleavage, and its inhibitory effect on PARP activity is correlated with the increase of DNA fragmentation in cancer cells. Moreover, *in silico* molecular docking studies were conducted to predict binding modes of compounds **10a** and **12a** in the AKT, ERK-1 and ERK-2, PARP and caspase-3 binding sites confirming the biological results. According to the analysis of the predicted binding modes and docking scores, the AKT protein seems to be the most interesting biological target to study the antiproliferative activity of the DAM. The docking study herein conducted can be used to guide further optimization of compounds **10a** and **12a.**

The structural adequacy, absence of cytotoxicity as well as druglikeness and favorable ADME profile, allow to suggest **10a**, **10b**, and **12a** are a new leads compounds in the study anticancer drugs. In general, these results could provide valuable insights to design new potent anticancer drugs based on the DAM scaffold.

## Data Availability

The data and materials in this article are available from the corresponding authors on reasonable request.

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
