# Peer review of "Novel Set of Diarylmethanes to Target Colorectal Cancer: Synthesis, In Vitro and In Silico Studies"

_biomolecules, 2022, doi:10.3390/biom13010054_

Round 1

Reviewer 1 Report

The paper entitled “Novel set of diarylmethanes to target colorectal cancer: synthesis, in vitro and in silico studies” presented for evaluation is scientifically interesting and well planned. After a few corrections and clarifications, it can be accepted into Biomolecules.

First of all, the authors should explain why they chose this type of cancer cells for research - it does not follow from the introduction. Did the authors do screening tests on other cancer cell lines and these turned out to be the ones most sensitive to the new diarylmethanes?

In my opinion, the values of the IC50 parameter for normal cell lines should be added to Table 1. The results presented in this way make it very difficult to compare cytotoxicity to cancer and normal cells lines.

What program the authors used for in silico research? It should be clearly indicated in the text, in the name Figure 14 and in the references.

Authors should standardize the notation of the CH3 group on diagrams. CH3 or Me, but the same everywhere.

The authors should repeat the HR MS analysis for substance 4, what is included in the S1 supplement is not a high-resolution mass analysis.

Reviewer 2 Report

·       All abbreviations should be defined when used for the first time.

·       Emphasise the novelty of the study.

·       What is the applicable value of the research, please discuss.

·       Abstract is chaotic. Please write it in a logical way.

·       There are many typos in the manuscript.

Section 2.6.2.:

·       Give full names of all cell lines studied, where are normal murine fibroblasts? This cell line should be added to methodology. Describe type of each cell line – cancer-non cancer; adherent, etc.

·       Justify why these cell lines were used for the study?

·       Give more detailed description on cell culturing (conditions, detaching, viability, etc.).

·       Line 642: did you remove MTT? Adding DMSO to MTT is incorrect, results are inaccurate. Such a procedure is incompatible with the ISO standard.

·       MTT is not a test for proliferation but for metabolic activity of cells. Therefore, the authors should not use the phrase 'antiproliferative effect' for this test, as this is not necessarily true. Antiproliferation is measured by another test such as BrDU. The authors also use viability and cytotoxicity interchangeably. This should be standardized. When referring to MTT, the form 'metabolic activity' is the most correct. It should be changed in the whole manuscript.

·       Figure 7 caption: what compounds?

·       Section 3.2.3. Please justify what was tested on these cells, how was it different from testing on cancer cells? This is very vaguely presented.

Round 2

Reviewer 2 Report

I have no more comments.